# Learning System Dynamics without Forgetting

**Xikun Zhang**
Generative AI Lab
College of Computing and Data Science
Nanyang Technological University
Singapore 639798
`xikun.zhang@ntu.edu.sg`

**Dongjin Song & Yushan Jiang**
Department of Computer Science and Engineering
University of Connecticut
Storrs, CT
USA
`{dongjin.song, yushan.jiang}@uconn.edu`

**Yixin Chen**
Washington University in St. Louis
St. Louis, MO
USA
`chen@cse.wustl.edu`

**Dacheng Tao**
Generative AI Lab
College of Computing and Data Science
Nanyang Technological University
Singapore 639798
`dacheng.tao@gmail.com`

## Abstract

Observation-based trajectory prediction for systems with unknown dynamics is essential in fields such as physics and biology. Most existing approaches are limited to learning within a single system with fixed dynamics patterns. However, many real-world applications require learning across systems with evolving dynamics patterns, a challenge that has been largely overlooked. To address this, we systematically investigate the problem of Continual Dynamics Learning (CDL), examining task configurations and evaluating the applicability of existing techniques, while identifying key challenges. In response, we propose the Mode-switching Graph ODE (MS-GODE) model, which integrates the strengths LG-ODE and sub-network learning with a mode-switching module, enabling efficient learning over varying dynamics. Moreover, we construct a novel benchmark of biological dynamic systems for CDL, Bio-CDL, featuring diverse systems with disparate dynamics and significantly enriching the research field of machine learning for dynamic systems. Our code available at `https://github.com/QueuQ/MS-GODE`.

## 1 Introduction

Scientific research often involves systems composed of interacting objects, such as multi-body systems in physics and cellular systems in biology, with their evolution governed by underlying dynamic rules. However, due to potentially unknown or incomplete dynamic rules or incomplete observations, deriving explicit equations to simulate system evolution can be extremely challenging. As a result, data-driven approaches based on machine learning have emerged as a promising solution for predicting the future trajectories of system states purely from observational data. For instance, the Interaction Network (IN) model (Battaglia et al., 2016) explicitly

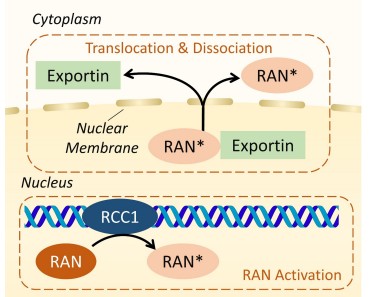

Figure 1: Illustration of the key components of one biological cellular system studied in our work: the RAN-regulated nucleocytoplasmic transport (Moore, 2013). Briefly speaking, this model depicts the translocation of cargo proteins (*Exportin*) via nuclear pores with the assistance of *RAN* proteins. *RAN* is first activated (*RAN\**) and then binds to cargo forming a complex. Next, the complex is translocated across the *nuclear membrane* into the cytoplasm with the assistance of *RAN*. Finally, *RAN* and *Exportin* are dissociated.

learns interactions between pairs of objects and has demonstrated superior performance in simulated physics systems, showcasing the potential of machine learning in studying physical system dynamics. IN has inspired many subsequent works (Kipf et al., 2018; Sanchez-Gonzalez et al., 2019; Huang et al., 2022; Liu et al., 2024). Later, to enable the modeling of incomplete and temporarily irregular system observations, ODE-based models (Huang et al., 2020; 2021) were proposed to learn the continuous dynamics of the systems.

Despite the success of these methods, existing approaches are often limited to learning within a single system with a fixed type of dynamics. However, in many real-world scenarios, system dynamics are subject to change over time. For example, in cellular systems (Figure 1), the dynamics of a set of variables are subject to change when the kinetic factors are altered. Besides, as a widely adopted task in the field, predicting the trajectories of n-body physical systems (Huang et al., 2020; Battaglia et al., 2016) may also require a model to learn over systems governed by different dynamics factors including interaction types (e.g., elastic force or electrostatic force) and interaction strengths (e.g., different amounts of charges on charged objects), as illustrated in Figure 7. In this work, we term these learning scenarios as continual dynamics learning (CDL), and for the first time formally formulate the setting of CDL. In CDL, continually learning from systems with varying dynamics may overwrite a model's knowledge encoded in the model weights and trigger the catastrophic forgetting problem. In other words, the model may only accurately predict the most recently observed system dynamics while failing on earlier systems. This phenomenon is empirically verified and reported in Section 4.6. Moreover, many real-world systems exhibit repeated dynamics, and mitigating the forgetting is actually crucial in a broad range of scenarios. For example, dynamics of many physics systems are controlled by environmental factors, e.g. temperatures (Huang et al., 2023). The values of these factors like temperatures typically oscillate within a certain range, therefore the dynamics of the systems will also repeat. This is also true in biological systems. Moreover, biological cellular systems also go through different phases of cell cycle. Catastrophic forgetting phenomenon has also been observed in other fields like computer vision (Van de Ven & Tolias, 2019; Wang et al., 2024) and graph learning (Zhang et al., 2024b), and different approaches have been proposed to facilitate the models in the continual learning setting. However, as demonstrated in our experiments (Section 4.3), most existing continual techniques, which are based on regularization or memory-replay (Yoon et al., 2017; Kirkpatrick et al., 2017; Lopez-Paz & Ranzato, 2017), are designed to accommodate the patterns of different tasks [1] within one set of model weights and fail to effectively alleviate the forgetting issue in the context of dynamics learning, especially when the consecutive systems contain different number of objects and exhibit significantly different dynamics patterns.

Targeting the challenge, we turn to the parameter-isolation based continual learning techniques, and propose a novel mode-switching graph ODE (MS-GODE) model, which can continually learn over varying system dynamics and automatically switch to the optimal mode during the test stage. MS-GODE consists of three major components: a prediction network serving as the backbone, a sub-network learning module for encoding the observed dynamics into masks, and a mode-switching module for switching the sub-network mode based on the observation. To support irregular and incomplete observational data in the practical scenario studied in our work, our backbone network is built based on LG-ODE model (Huang et al., 2020), which has a Variational AutoEncoder (VAE) (Kingma & Welling, 2013) structure facilitated by ODE-based prediction (Rubanova et al., 2019). The basic workflow of MS-GODE is as follow: Given the observational data, an encoder network first encodes the data into latent states, upon which an ODE-based generator predicts the future system trajectories within the latent space. Finally, a decoder network maps the predicted latent states back into the data space. Upon this framework , we adopt the sub-network learning strategy (Wortsman et al., 2020; Ramanujan et al., 2020; Zhou et al., 2019), which fixes the model weights after initialization and optimize a unique binary mask over the parameters for each system during training. In this way, unlike standard training strategy that encodes data patterns solely in one set of model weights, MS-GODE encodes different types of dynamics in different sub-networks by the collaboration between the binary masks and the fixed-weight backbone network. In the test stage, the model will be switched to the optimal mode by the switching module via selecting the the most suitable mask that can most accurately reconstruct the given observation. Moreover, targeting that the existing entropy-based mask selection technique is only applicable to classification tasks, we further develop the novel observation reconstruction based mask selection strategy in the mode-switching

---

[1]In our work, a task refers to a system with a specific dynamics patterns. While in other fields, *e.g.* Computer Vision, a task could refer to certain categories of images.

module. In this way, despite the significant difference between consecutive systems, catastrophic forgetting is avoided.

Besides innovatively formulating the CDL setting and the technical contribution of an effective model in CDL scenario, we have also created a novel dynamic system benchmark, *Bio-CDL* consisting of biological cellular systems based on the VCell platform (Schaff et al., 1997; Cowan et al., 2012; Blinov et al., 2017). Compared to the widely adopted simulated physics systems, cellular systems contain heterogeneous objects and interactions, offering richer and more challenging patterns of system dynamics. Therefore, Bio-CDL will significantly enhance the research of machine learning-based system dynamics prediction. In experiments, we thoroughly investigate the influence of the system sequence configuration on model performance in CDL with both the widely adopted physics systems and our newly constructed BioCDL, which demonstrates the advantage of MS-GODE over existing state-of-the-art techniques.

## 2 RELATED WORKS

### 2.1 LEARNING BASED DYNAMIC SYSTEM PREDICTION

In recent years, graph neural networks (GNNs) have been proven to be promising in modeling and predicting the complex evolution of systems consisting of interacting objects (Battaglia et al., 2016; Kipf et al., 2018; Sanchez-Gonzalez et al., 2019; Huang et al., 2022; Liu et al., 2024; Luo et al., 2024b; Zhang et al., 2019; Luo et al., 2024a; Wu et al., 2021; Luo et al., 2025; Jiang et al., 2024; Guo et al., 2024; Zhang et al., 2020b; Shen et al., 2024; Yin et al., 2024; Chen et al., 2025). This was firstly demonstrated by (Battaglia et al., 2016) with Interaction Network (IN), which iteratively infers the effects of the pair-wise interactions within a system and predicts the changes of the system states. Following IN, (Kipf et al., 2018) proposed neural relational inference (NRI) to predict systems consisting of objects with unknown relationships. (Mrowca et al., 2018) proposed the Hierarchical Relation Network (HRN) that extends the predictions to systems consisting of deformable objects. (Sanchez-Gonzalez et al., 2019)

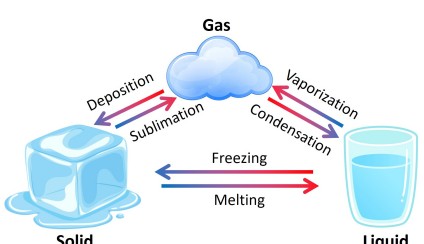

Figure 2: A molecule system may enter different phases and exhibit different dynamics as environmental factors (*e.g.* temperature) change. Molecules in solid state can only vibrate at fixed locations because of the strong interaction between them. Upon entering the liquid state, the interaction strength decreases and molecules can move around. In gas state, molecules move more freely with little molecule-wise interaction.

proposed the Hamiltonian ODE graph network (HOGN), which injects Hamiltonian mechanics into the model as a physically informed inductive bias. Later, to better consider the intrinsic symmetry of the target systems, GNNs with different invariance and equivariance are proposed (Satorras et al., 2021; Huang et al., 2022; Han et al., 2022; Brandstetter et al., 2021; Wu et al., 2023; Liu et al., 2024). To better capture the complex system interactions, High-order graph ODE (HOPE) (Luo et al., 2023) innovatively incorporates information from high-order spatial neighborhood and high-order derivatives into dynamical system modeling. Considering that the observation of real-world systems may be incomplete and irregular samples, (Huang et al., 2020) proposed LG-ODE, which is capable of generating continuous system dynamics based on the latent ordinary differential equations. Later, Coupled Graph ODE (CG-ODE) (Huang et al., 2021) was proposed to apply ODE-based modeling to both node features and interactions. Similar ideas are also adopted in other time series research (Rubanova et al., 2019). By separating the commonalities intrinsic to the systems and the environmental factors causing the dynamics shift, Generalized Graph Ordinary Differential Equations (GG-ODE) (Huang et al., 2023) improves the generalization across systems in different environments. Similarly, Prototypical Graph ODE (PGODE) (Luo et al.) disentangles object states and system states to independently model their influence and improve the generalization capability. Disentangled Intervention-based Dynamic graph Attention networks (DIDA) (Zhang et al., 2022c) disentangles the invariant and variant patterns in dynamic graphs, and leverages the invariant patterns to ensure a stable prediction performance under spatio-temporal distribution shift. Context-attended Graph ODE (CARE) (Luo et al., 2024c) models the continuously varying environmental factors with a context variable, which is leveraged to better predict the system evolution with temporal environmental variation. Online Relational Inference (ORI) (Kang et al., 2024) models the relationship as trainable parameters, which is accompanied by AdaRelation for the online setting. Despite the substantial contributions these methods have made to dynamic system prediction, they have been limited to learning a single system with fixed dynamics. As one of the two major components of our MS-GODE,

the backbone model for dynamics system prediction is mainly built upon the LG-ODE framework (Huang et al., 2020). Different from the other approaches that are typically limited to data with regular intervals or complete observation at each time stamp, LG-ODE supports irregular and incomplete observations, which is essential to the applications studied in our work.

## 2.2 CONTINUAL LEARNING & MASKED NETWORKS

Existing continual learning methods can be categorized into three types (Parisi et al., 2019; Van de Ven & Tolias, 2019; De Lange et al., 2021; Zhang et al., 2022a; Konishi et al., 2023; Zhang et al., 2024a; 2023b; 2022b; Fionda et al., 2023). Regularization-based methods slow down the adaption of important model parameters via regularization terms, so that the forgetting problem is alleviated (Wu et al., 2024; Goswami et al., 2023). For example, Elastic Weight Consolidation (EWC) (Kirkpatrick et al., 2017) and Memory Aware Synapses (MAS) (Aljundi et al., 2018) estimate the importance of the model parameters to the learned tasks, and add penalty terms to slow down the update rate of the parameters that are important to the previously learned tasks. Second, experience replay-based methods replay the representative data stored from previous tasks to the model when learning new tasks to prevent forgetting (Liang & Li, 2023; Rolnick et al., 2019; Rebuffi et al., 2017; Prabhu et al., 2020). For example, Gradient Episodic Memory (GEM) (Lopez-Paz & Ranzato, 2017) leverages the gradients computed based on the buffered data to modify the gradients for learning the current task and avoid the negative interference between learning different tasks. Finally, parameter isolation-based methods gradually introduce new parameters to the model for new tasks to prevent the parameters that are important to previous tasks (Qiao et al., 2023; Yoon et al., 2017). For example, Progressive Neural Network (PNN) (Rusu et al., 2016) allocates new network branches for new tasks, such that the learning on new tasks does not modify the parameters encoding knowledge of the old tasks. Our proposed MS-GODE also belongs to the parameter isolation-based methods, and is related to subnetwork-based ones (Wortsman et al., 2020; Kang et al., 2022; Zhou et al., 2019) and the edge-popup algorithm (Ramanujan et al., 2020). SupSup studies continual learning for classification tasks with an output entropy-based mask selection, which is not applicable to our task. Edge-popup algorithm provides a simple yet efficient strategy to select a sub-network, and is adopted by us to optimize the binary masks over the model parameters.

## 3 LEARNING SYSTEM DYNAMICS WITHOUT FORGETTING

### 3.1 PRELIMINARIES

In CDL, a model is required to sequentially learn on multiple systems. A system is composed of multiple interacting objects, and is naturally represented as a graph $\mathcal{G} = \{\mathbb{V}, \mathbb{E}\}$. $\{\mathbb{V}$ is the node set denoting the objects of the system, and $\mathbb{E}$ is the edge set containing the information of the relation and interaction between the objects. Based on $\mathbb{E}$, the spatial neighbors of a node $v$ is defined as $\mathcal{N}_s(v) = \{u | e_{u,v} \in \mathbb{E}\}$. Each object node $v$ is accompanied by observational data containing the observed states at certain time steps $\mathbb{X}_v = \{\mathbf{x}_v^t | t \in \mathbb{T}_v\}$, *i.e.* the trajectory of the system evolution. With a system structured as a graph, its trajectory is naturally a spatial-temporal graph, in which each node is an observed state $\mathbf{x}_v^t$. In the following, we will refer to $\mathbf{x}_v^t$ as a 'state'. In a multi-body system, the trajectory records the 3D locations of the particles over time. While in other systems, *e.g.* cellular systems, a state could be the amount of a certain substance instead of locations, and a trajectory records the states at different time stamps. The set $\mathbb{T}_v$ contains the time steps (real numbers) when the states of $v$ are observed and can vary across different objects. For the prediction task, the observations lie within a certain period, *i.e.* $\bigcup_{v \in \mathbb{V}} \mathbb{T}_v \in [t_0, t_1]$, and the task is to predict the system states at future time steps beyond $t_1$. We denote the future time steps to predict for an object $v$ as $\mathbb{T}_v^{pred}$. In CDL, different systems in a sequence may exhibit different dynamics and contain different objects (*i.e.* $\mathbb{V}$).

### 3.2 FRAMEWORK OVERVIEW

In this subsection, we provide a high-level introduction to the workflow of MS-GODE (Figure 3), while the details of each component are provided in the following subsections.

Overall, MS-GODE consists of three core components: 1) The backbone network; 2) The sub-network learning module; 3) The mode switching module. The backbone network consists of: 1) an encoder network $\text{Enc}(\cdot; \theta_E)$ parameterized by the parameters $\theta_E$ for encoding the trajectories into the latent space; 2) An ODE-based generator $\text{Gen}(\cdot; \theta_G)$ parameterized by $\theta_G$ for predicting the future trajectories within the latent space; 3) A decoder network $\text{Dec}(\cdot; \theta_D)$ parameterized by $\theta_D$ for transforming the predicted latent states back into the data space. Within a standard learning

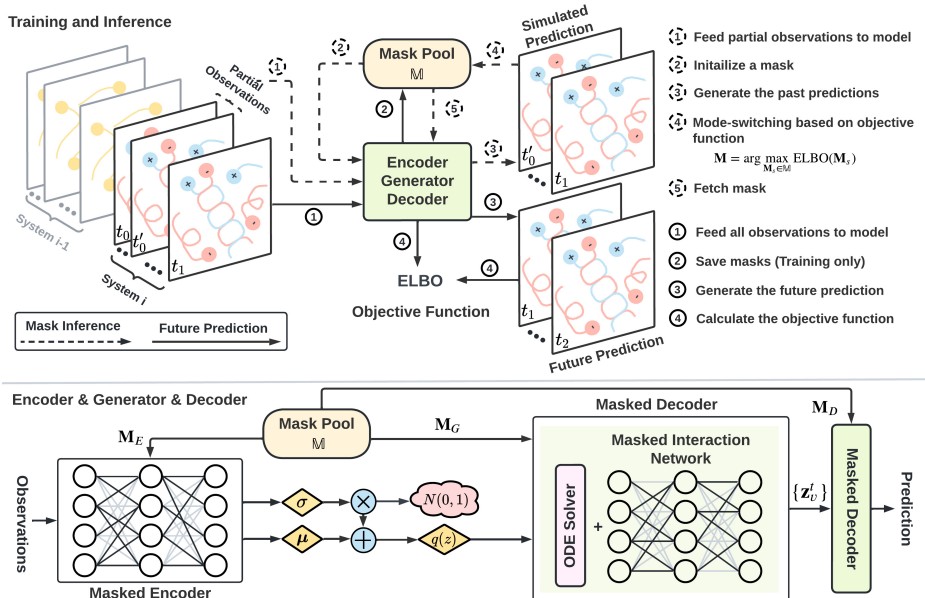

Figure 3: The upper part illustrates the workflow of MS-GODE during mask selection and inference, which are denoted by dashed and solid lines. During mask selection, the observation is split into two parts, and the first part is fed into the model for selecting the mask that can best predict the second part. During inference, the entire observation is fed into the model for prediction of future states. The lower part demonstrates the structure of the masked encoder, masked generator, and masked decoder. Different components fetch their corresponding mask from the mask pool and apply the mask onto the parameters.

scheme, the model will be trained by updating the parameters $\{\theta_E, \theta_G, \theta_D\}$. However, as introduced above, when continually training the model on multiple systems with different dynamics, this learning scheme will bias the model towards the most recently observed system, causing the catastrophic forgetting problem. In this work, inspired by the recent advances in sub-network learning (Ramanujan et al., 2020; Wortsman et al., 2020), we propose to avoid direct optimization of the parameters $\{\theta_E, \theta_G, \theta_D\}$. Instead, we train the model by optimizing the connection topology of the backbone model and encode the dynamics of each system into a sub-network. This is equivalent to encoding each system-specific dynamics pattern into a binary mask overlaying the shared parameters with fixed values. In this way, the interference between learning on different systems with different dynamics, *i.e.* the forgetting problem, can be avoided. Moreover, this approach is also memory efficient since the space for storing the binary masks is negligible. With the system-specific masks, the three model components are formulated as:

$$\text{Enc}(\cdot; \theta_E \odot \mathbf{M}_E^s); \quad \text{Gen}(\cdot; \theta_G \odot \mathbf{M}_G^s); \quad \text{Dec}(\cdot; \theta_D \odot \mathbf{M}_D^s), \tag{1}$$

where the superscript $s$ is the system index. The details of the binary mask optimization during training and mask selection during testing are provided in Section 3.5. In the following subsections, all parameters are subsets of $\theta_E$, $\theta_G$, or $\theta_D$ and are controlled by the corresponding masks. For example, $\mathbf{W}_{msg}$ in Section 3.3 is part of the encoder parameters $\theta_E$ and is under the control of $\mathbf{M}_E^s$.

### 3.3 MASKED ENCODER NETWORK

As the first component of the model, the masked encoder network serves to encode the dynamics pattern in the observational data into the latent space. As introduced in Section 3.1, the trajectory of a system is a spatial-temporal graph. Therefore, the encoder is constructed as an attention-based spatial-temporal graph neural network framework (ST-GNN) (Huang et al., 2020; Hu et al., 2022; Huang et al., 2021; Zhang et al., 2020a). Originally, the graph attention network (GAT) (Veličković et al., 2017) is designed to aggregate information over the spatially neighboring nodes. On spatial-temporal graphs, the information aggregation is extended to both spatial and temporal neighboring nodes. Such a spatial-temporal neighborhood of a state $\mathbf{x}_v^t$ is defined as the states of the spatially connected nodes within a specified temporal window $\delta_{window}$,

$$\mathcal{N}_{st}(\mathbf{x}_v^t) := \{\mathbf{x}_w^q | e_{u,v} \in \mathbb{E} \quad and \quad |q - t| < \delta_{window}\}. \tag{2}$$

Then, iterative message passing is conducted over the spatial-temporal edges to update the representation of each state. The update of the hidden representation of a state $\mathbf{x}_v^t$ at the $l$-the layer of the ST-GNN is formulated as,

$$\mathbf{h}_{v,t}^l = \mathbf{h}_{v,t}^{l-1} + \sigma\Big(\sum_{x_u^q \in \mathcal{N}_{st}(\mathbf{x}_v^t)} \mathrm{a}(\mathbf{h}_{v,t}^{l-1}, \mathbf{h}_{u,q}^{l-1}, q-t) \cdot \mathbf{W}_{msg} \cdot \mathrm{msg}(\mathbf{h}_{u,t}^{l-1}, q-t)\Big), \tag{3}$$

where $\mathrm{a}(\cdot, \cdot)$ calculates the attention score between a pair of states, and $\mathrm{msg}(\cdot, \cdot)$ is the message function commonly adopted in GNNs (Gilmer et al., 2017). However, different from the message function in typical GNNs, the temporal relationship between the states is a crucial part to understand the system dynamics. Therefore, we follow the strategy in LG-ODE (Huang et al., 2020) to incorporate the temporal distance between the states in the message function and attention function,

$$\mathrm{msg}(\mathbf{h}_{u,t}^{l-1}, q-t)) := \sigma(\mathbf{W}_{tmp} \cdot \mathrm{concat}(\mathbf{h}_{u,t}^{l-1}, q-t)) + \mathrm{TE}(q-t), \tag{4}$$

$$\mathrm{a}(\mathbf{h}_{v,t}^{l-1}, \mathbf{h}_{u,q}^{l-1}, q-t) := \frac{\exp(\mathrm{msg}(\mathbf{h}_{u,q}^{l-1}, q-t))^T \cdot \mathbf{h}_{v,t}^{l-1})}{\sum_{x_w^p \in \mathcal{N}_{st}(\mathbf{x}_v^t)} \exp(\mathrm{msg}(\mathbf{h}_{w,p}^{l-1}, p-t))^T \cdot \mathbf{h}_{v,t}^{l-1})}, \tag{5}$$

where $\mathrm{TE}(\cdot)$ is a temporal position encoding developed based on the position encoding in Transformer (Vaswani et al., 2017) for incorporating the temporal information into the representation. Finally, for subsequent state prediction, the state representations are averaged over the temporal dimension,

$$\mathbf{h}_{final}^v = \frac{1}{|\mathbb{T}_v|} \sum_{t \in \mathbb{T}_v} \sigma(\bar{\mathbf{h}}_v^T \cdot \mathrm{msg}(\mathbf{h}_{v,t}^L)) \cdot \mathrm{msg}(\mathbf{h}_{v,t}^L, t-t_0), \tag{6}$$

where $t_0$ denotes the starting time of all states in the observational data (Section 3.1), and the average term $\bar{\mathbf{h}}_v$ of each node $v$ is a weighted summation over the representations at all time steps,

$$\bar{\mathbf{h}}_v = \sigma\Big(\frac{1}{|\mathbb{T}_v|} \sum_{t \in \mathbb{T}_v} \mathrm{msg}(\mathbf{h}_{v,t}^L, t-t_0) \cdot \mathbf{W}_{avg}\Big). \tag{7}$$

## 3.4 MASKED ODE-BASED GENERATOR

ODE-based generator (Huang et al., 2020; Chen et al., 2018; Rubanova et al., 2019) ensures that the model can handle observations with irregular temporal intervals and incomplete states, as well as predict future states at any time denoted by real numbers. Specifically, the trajectory prediction is formulated as solving an ODE initial value problem (IVP), where the initial values of the objects ($\{z_v^{t_1} | v \in \mathbb{V}\}$) are generated from the final representation of the states ($\{\mathbf{h}_{final}^v | v \in \mathbb{V}\}$, Section 3.3). Mathematically, the procedure of predicting the future trajectory of a system $s$ is formulated as,

$$z_v^{t_1} \sim \mathrm{p}(z_v^{t_1}), v \in \mathbb{V} \tag{8}$$

$$\{z_v^\tau | v \in \mathbb{V}, \tau \in \mathbb{T}_v^{pred}\} = \mathrm{Gen}(\{z_v^{t_1} | v \in \mathbb{V}\}, \{\mathbb{T}_v^{pred} | v \in \mathbb{V}\}; \theta_G \odot \mathbf{M}_G^s)), \tag{9}$$

To estimate the posterior distribution $q(\{z_v^{t_1} | v \in \mathbb{V}\} | \{\mathbb{X}_v | v \in \mathbb{V}\})$ based on the observation (*i.e.* $\{\mathbb{X}_v | v \in \mathbb{V}\}$), the distribution is assumed to be Gaussian. Then the mean $\mu_v$ and standard deviation $\sigma_v$ are generated from $\{\mathbf{h}_{final}^v | v \in \mathbb{V}\}$ with a multi-layer perceptron (MLP),

$$q(z_v^{t_1} | \{\mathbb{X}_v | v \in \mathbb{V}\}) = \mathcal{N}(\mu_v, \sigma_v) = \mathcal{N}(\mathrm{mlp}(\mathbf{h}_{final}^v; \theta_G \odot \mathbf{M}_G^s)), v \in \mathbb{V}. \tag{10}$$

As noted in Section 3.2, the parameters of $\mathrm{mlp}(\cdot)$ and $\mathrm{F}_{int}(\cdot)$ are part of $\theta_G$ and controlled by $\mathbf{M}_G^s$.

Based on the approximate posterior distribution $q(\{z_v^{t_1} | v \in \mathbb{V}\} | \{\mathbb{X}_v | v \in \mathbb{V}\})$, we sample an initial state for each object, upon which the ODE solver will be applied for generating the predicted states in the latent space. The dynamics of each object in the system are governed by its interaction with all the other objects. Therefore, the core part of the ODE-based generator is a trainable interaction network that encodes the dynamics in the form of the derivative of each $\mathbf{z}_v$,

$$\frac{\mathrm{d}\mathbf{z}_v}{\mathrm{d}t}\Big|_{t=t'} = \mathrm{F}_{int}(\{\mathbf{z}_u^{t'} | u \in \mathcal{N}_s(v)\}; \theta_G \odot \mathbf{M}_G), \tag{11}$$

where the function $\mathrm{F}_{int}(\cdot)$ parameterized by $\theta_I$ predicts the dynamics (derivative) of each object $v$ based on all the other objects interacting with $v$ (*i.e.* $\mathcal{N}_s(v)$ defined in Section 3.1), and $t'$ denotes any possible future time. Note that $\mathcal{N}_s(v)$ only contains the spatial neighbors and is different from

$\mathcal{N}_{st}(\cdot)$, because the object-wise interaction at a certain time $t'$ is not dependent on system states at other times. For example, in a charged particle system governed by electrostatic force, the force between a pair of particles at $t'$ is solely determined by the relative positions (*i.e.* the states) of the particles (Figure 7) at $t'$. In our work, we adopt the Neural Relational Inference (NRI) (Kipf et al., 2018) as the function $\mathrm{F}_{int}(\cdot)$. Based on $\mathrm{F}_{int}(\cdot)$, the future state of the system at an arbitrary future time $t_2$ can be obtained via an integration

$$\mathbf{z}_v^{t_2} = \mathbf{z}_v^{t_1} + \int_{t_1}^{t_2} \frac{\mathrm{d}\mathbf{z}_v}{\mathrm{d}t} \mathrm{d}t = \mathbf{z}_v^{t_1} + \int_{t_1}^{t_2} \mathrm{F}_{int}(\{\mathbf{z}_u^t | u \in \mathcal{N}_s(v)\}; \theta_G \odot \mathbf{M}_G) \mathrm{d}t, \tag{12}$$

which can be solved numerically by mature ODE solvers, *e.g.* Runge-Kutta method. After obtaining the latent representations of the states at the future time steps ($\{\mathbf{z}_v^t | t \in \mathbb{T}_{future}, v \in \mathbb{V}\}$), the predictions are generated via a masked decoder network that projects the latent representations back into the data space

$$\mathbf{y}_v^t = \mathrm{Dec}(\mathbf{z}_v^t; \theta_D \odot \mathbf{M}_D), t \in \mathbb{T}_{future}, v \in \mathbb{V}. \tag{13}$$

In our work, $\mathrm{Dec}(\cdot; \theta_D)$ is instantiated as a multi-layer perceptron (MLP).

### 3.5 SUB-NETWORK LEARNING

MS-GODE is trained by maximizing the Evidence Lower Bound (ELBO). Denoting the concatenation of all the initial latent states ($\{\mathbf{z}_v^{t_1} | v \in \mathbb{V}\}$) as $\mathbf{Z}_{\mathbb{V}}^{t_1}$, the ELBO is formulated as

$$\mathrm{ELBO}(\mathbf{M}^s) = \mathbb{E}_{\mathbf{Z}_{\mathbb{V}}^{t_1} \sim q(z_v^{t_1} | \{\mathbb{X}_v | v \in \mathbb{V}\})} [\log(p(\{\mathbf{x}_v^t | t \in \mathbb{T}_{future}, v \in \mathbb{V}\}))] \tag{14}$$

$$- \mathbf{KL}[q(\mathbf{Z}_{\mathbb{V}}^{t_1} | \{\mathbb{X}_v | v \in \mathbb{V}\})) || p(\mathbf{Z}_{\mathbb{V}}^{t_1})]. \tag{15}$$

where $p(\mathbf{Z}_{\mathbb{V}}^{t_1})$ denotes the prior distribution of $\mathbf{Z}_{\mathbb{V}}^{t_1}$, which is typically chosen as standard Gaussian. $\mathbf{M}^s$ denotes the union of all masks over different modules of the framework (*i.e.* $\mathbf{M}^s = \{\mathbf{M}_E^s, \mathbf{M}_I^s, \mathbf{M}_P^s\}$). In our model, the parameters ($\theta_E, \theta_I$, and $\theta_P$) will be fixed, and the ELBO will be maximized by optimizing the binary masks $\mathbf{M}^s$ overlaying the parameters via the Edge-popup algorithm (Ramanujan et al., 2020). After learning each system, the obtained mask is added into a mask pool $\mathbb{M}$ to be used in testing.

### 3.6 MODE-SWITCHING

During testing, MS-GODE will be evaluated on a sequence of systems exhibiting diverse dynamics patterns, and it will automatically switch to the optimal mode that ensures the highest accuracy. This is achieved by applying the most suitable mask based on the performance of reconstructing part of the given observations. To obtain the most suitable mask, a given observation from $[t_0, t_1]$ is first split it into two periods $[t_0, \frac{t_0+t_1}{2}]$ and $[\frac{t_0+t_1}{2}, t_1]$. Then, the first half is fed into the model and the correct mask can be chosen by selecting the one that can reconstruct the second half with the lowest error.

## 4 EXPERIMENTS

In this section, we aim to answer the following questions. 1. How to properly configure MS-GODE for optimal performance? 2. How would the configuration of the system sequence influence the performance? 3. How is the performance of the existing CL techniques? 4. Can MS-GODE outperform the baselines?

### 4.1 EXPERIMENTAL SYSTEMS

In experiments, we adopt physics and biological cellular systems. A detailed introduction to the system sequence construction is provided in Appendix A.1.

**Simulated physics systems** are commonly adopted to evaluate the machine learning models in the task of learning system dynamics (Liu et al., 2024; Battaglia et al., 2016; Huang et al., 2020). The physics systems adopted in this work include spring-connected particles and charged particles (Figure 7). We carefully adjust the system configuration and construct 3 system sequences with different levels of dynamics shift (Appendix A.1).

**Biological cellular systems** are innovatively introduced in this work based on Virtual Cell platform (Schaff et al., 1997; Cowan et al., 2012; Blinov et al., 2017). Currently, Bio-CDL includes rule-based model of EGFR and compartment rule based model of translocation based on the Ran protein

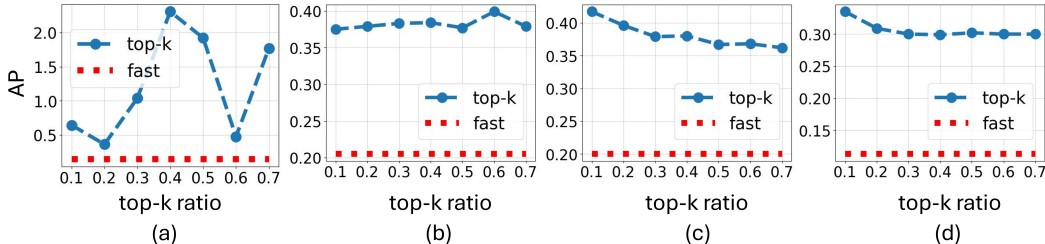

Figure 4: Performance comparison among different strategies to binarize the mask values. (a) Comparison over the cellular system sequence $\text{EGFR}_1 \rightarrow \text{Ran}_1 \rightarrow \text{EGFR}_2 \rightarrow \text{Ran}_2$. (b)(c)(d) Comparison over different physics system sequences. Blue line denotes the performance of top-k selection with different thresholds. Red line demonstrates the performance of using fast selection.

(GTPase). In experiments, we adjust the coefficients of the models to construct a sequence containing 2 EGFR and 2 Ran systems interleaved with each other ($\text{EGFR}_1 \rightarrow \text{Ran}_1 \rightarrow \text{EGFR}_2 \rightarrow \text{Ran}_2$). Full details are provided in Appendix A.1.2.

## 4.2 EXPERIMENTAL SETUPS & EVALUATION

**Model evaluation in CDL.** The models in this work learn sequentially on multiple systems under the continual learning setting, thus the evaluation is significantly different from standard learning settings. After learning each new task, the model is tested on all learned tasks and the results form a performance matrix $M^p \in \mathbb{R}^{N \times N}$, where $M^p_{i,j}$ denotes the performance on the $j$-system after learning from the 1-st to the $i$-th system, and $N$ is the number of systems in the sequence. In our experiments, each entry of $M^p$ is a mean square error (MSE) evaluating the performance on a single system. To evaluate the performance over all systems, average performance (AP) can be calculated. For example, $\frac{\sum_{j=1}^{N} M^p_{N,j}}{N}$ is the average performance after learning the entire sequence with $N$ tasks. Similarly, average forgetting (AF) can be calculated as $\frac{\sum_{j=1}^{N-1} M^p_{N,j} - M^p_{j,j}}{N-1}$. More details on model evaluation can be found in Appendix A.4. The baseline are configured by combining existing continual learning techniques with the LG-ODE (Huang et al., 2020) model. Other dynamics learning models were excluded because they typically don't support the practical setting with irregular and incomplete observation studied in this work. All experiments are repeated 5 times on a Nvidia Titan Xp GPU. The results are reported with average and standard deviations.

**Baselines & model settings.** We adopt state-of-the-art baselines including the performance upper (joint training) and lower bounds (fine-tune) in experiments. The state-of-the-art baselines adopted in this work include Elastic Weight Consolidation (EWC) (Kirkpatrick et al., 2017) based on regularization, Learning without Forgetting (LwF) (Li & Hoiem, 2017) based on knowledge distillation, Gradient Episodic Memory (GEM) (Lopez-Paz & Ranzato, 2017) based on both memory replay and regularization, Bias-Correction (Chrysakis & Moens, 2023) based Memory Replay (BCMR) that integrates the idea of data sampling bias correction into memory replay, and Scheduled Data Prior (SDP) (Koh et al., 2023) that adopts a data-driven approach to balance the contribution of past and current data. Besides, joint training and fine-tuning are commonly adopted in continual learning works. Joint training trains the model jointly over all systems, which does not follow the continual learning setting. Fine-tune directly trains the model incrementally on new systems without any continual learning technique. As revealed by Ramanujan et al. (2020), for learning subnetworks, pre-trained models are the most suitable for serving as the backbone. However, unlike the Computer Vision (CV) tasks studied by Ramanujan et al. (2020), pre-trained models are not readily available in the context of dynamics system modeling. For example, Seifner et al. (2024) provides a promising pre-trained model with valuable insights into pre-training for dynamical systems, but the model is still limited to interpolation tasks. However, we will investigate how to adapt the provided model to extrapolation tasks once their code and model are released. Therefore, we adopt the random initialization strategy, which is shown by Ramanujan et al. (2020) to be less effective but could be comparable to pre-trained models with a proper ratio of remaining parameters, which is thoroughly investigated in our experiments reported in Section 4.3 (Figure 4). More details of experimental settings and baselines and experimental settings are provided in Appendix A.1 A.3.

## 4.3 MODEL CONFIGURATION AND PERFORMANCE (RQ1)

Table 1: Performance comparisons on physics system sequences (↓ lower means better).

| Method | Seq1: Low-level dynamics shift | | Seq2: Mid-level dynamics shift | | Seq3: High-level dynamics shift | |
|---|---|---|---|---|---|---|
| | AP ↓ | AF ↓ | AP ↓ | AF /% ↓ | AP ↓ | AF /% ↓ |
| Fine-tune | 0.369±0.027 | 0.115±0.016 | 0.391±0.044 | 0.314±0.030 | 0.258±0.025 | 0.086±0.037 |
| EWC 2017 | 0.208±0.015 | -0.007±0.019 | 0.227±0.038 | 0.008±0.022 | 0.148±0.011 | 0.008±0.017 |
| GEM 2017 | 0.251±0.037 | 0.079±0.020 | 0.379±0.023 | 0.302±0.030 | 0.163±0.037 | -0.091±0.033 |
| LwF 2017 | 0.258±0.011 | 0.104±0.042 | 0.363±0.039 | 0.312±0.042 | 0.130±0.025 | 0.016±0.032 |
| BCMR 2023 | 0.284±0.017 | -0.006±0.031 | 0.298±0.028 | -0.001±0.033 | 0.233±0.017 | 0.027±0.023 |
| SDP 2023 | 0.352±0.021 | 0.121±0.018 | 0.303±0.026 | 0.352±0.058 | 0.213±0.035 | 0.024±0.045 |
| Joint | 0.194±0.006 | - | 0.186±0.015 | - | 0.116±0.009 | - |
| **Ours** | **0.200±0.003** | 0.002±0.004 | **0.204±0.005** | -0.001±0.001 | **0.113±0.001** | -0.000±0.000 |

When optimizing the system-specific masks using the edge-popup algorithm (Ramanujan et al., 2020) (Appendix A.2), each entry of the mask is assigned with a continuous value for gradient descent after backpropagation. During inference, different strategies can be adopted to transform the continuous scores into binary values. In our experiments, we tested both 'fast selection' and 'top-k selection' with different thresholds. 'Fast selection' sets all entries with positive score values into '1's and the other entries into '0's. 'Top-k selection' first ranks the score values and sets a specified ratio of entries with the largest values into '1's. In Figure 4, we show the performance of different strategies. Overall, 'top-k selection' is inferior to 'fast selection'. This is potentially because 'fast selection' does not limit the number of selected entries, therefore allowing more flexibility for optimization. We also observe that the performance of 'top-k' selection is more sensitive on the cellular systems compared to the physics systems, indicating that the cellular systems have higher optimization difficulty for sub-network (binary mask) learning over system sequences.

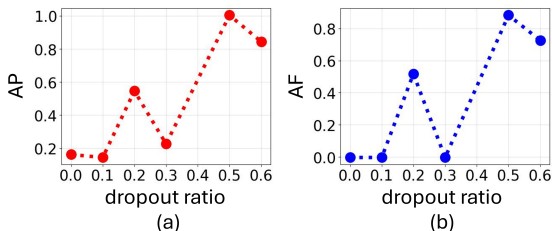

Figure 5: AP (a) and AF (b) of MS-GODE with different dropout rate.

Second, sub-network learning will deactivate some neurons in the model, which resembles the dropout mechanism widely adopted in machine learning models and may cause the model to be over-sparsified. Therefore, we investigate the influence of dropout rate in MS-GODE. From Figure 5, we can see that a smaller dropout rate results in lower error (better performance). This corroborates our hypothesis that the mask selection mechanism and dropout complement each other, and the dropout rate should be decreased when the masking strategy is adopted.

## 4.4 SEQUENCE CONFIGURATION AND PERFORMANCE (RQ2)

In this subsection, we investigate the influence of system sequence configuration on the learning difficulty and performance. Specifically, we construct 3 physics system sequences with increasing level of dynamics shift (Appendix A.1.1). Sequence 1 (low-level dynamics shift) consists of 8 spring connected particle systems, in which consecutive systems are only different in one system coefficient. Sequence 2 (mid-level dynamics shift) is constructed to have a higher level of dynamics shift by simultaneously varying 2 system coefficients. Finally, sequence 3 (high-level dynamics shift) is constructed by interleaving spring-connected particle systems and charged particle systems with disparate dynamics. As shown in Table 1, in terms of both AP and AF, most methods, including MS-GODE, obtain similar performance over Sequence 1 and 2, and obtain better performance on Sequence 3. For MS-GODE, since the systems with more diverse dynamics are easier to distinguish for selecting the masks during inference. For all the methods in general, two possible cases are: 1. The model is not well trained on any system (it has not encoded much information), therefore it has nothing to forget. 2. The model is well trained and can maintain enough information from each system. However, the lower AP (low MSE) obtained on the system sequence with high-level dynamics shift indicates that the model has well adapted to each system, i,e, case 1 is not true. When case 2 holds, it indicates that the model's parameters are sufficiently updated to fit each system. When learning on one system, two possible cases are: 1. All parameters are modified uniformly. 2. A subset of parameters are modified more than the others. When the baseline is Fine-tune, if all the parameters are uniformly modified, it would be almost impossible to maintain the performance on previous systems. Therefore, the potential case is that each system relies more on a subset of the

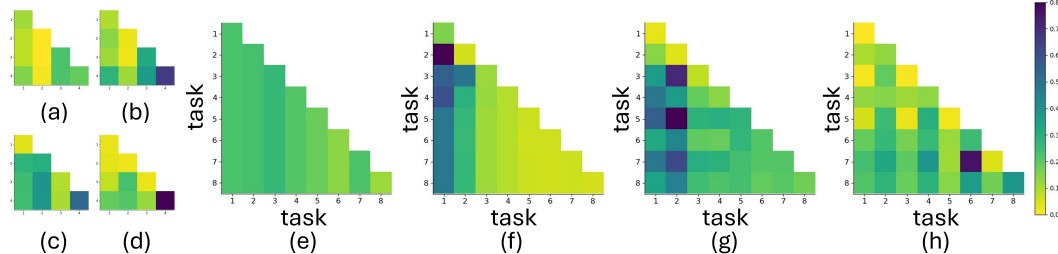

Figure 6: Visualization of performance matrices. (a), (b), (c), (d) corresponds to MS-GODE, fine-tune, EWC, and LwF on the cellular system sequence. (e), (f), (g), (h) are the performance of MS-GODE, EWC, fine-tune, and GEM, on the physics systems. Lower values indicate better performance.

parameters, and systems with larger difference in dynamics may lead to less overlap in the subsets of parameters they rely on.

### 4.5 COMPARISONS WITH STATE-OF-THE-ARTS (RQ3,4)

In this subsection, we compare MS-GODE with state-of-the-arts methods including the joint training. The experiments are conducted on both physics systems (Table 1) and cellular systems (Table 2). Besides, by comparing the results across different sequences in the two tables, we find that the sequences with gradual dynamics shift (Sequence 1,2 of physics systems) are more difficult to learn than the sequences with abrupt dynamics shift (Sequence 3 of the physics systems and the cellular system).The advantages of MS-GODE over the baselines mainly come from two perspectives. First, since the learning of the subnetworks are learned in a completely

Table 2: Performance comparisons on biological cellular systems ($\downarrow$ lower means better).

| Method | $\text{EGFR}_1 \rightarrow \text{Ran}_1 \rightarrow \text{EGFR}_2 \rightarrow \text{Ran}_2$ | |
|---|---|---|
| | AP $\downarrow$ | AF $\downarrow$ |
| Fine-tune | 0.355±0.089 | 0.226±0.037 |
| EWC 2017 | 0.312±0.028 | -0.013±0.019 |
| GEM 2017 | 0.316±0.083 | 0.352±0.109 |
| LwF 2017 | 0.330±0.036 | 0.349±0.046 |
| BCMR 2023 | 0.149±0.025 | 0.013±0.044 |
| SDP 2023 | 0.197±0.025 | 0.167±0.045 |
| Joint | 0.055±<0.001 | - |
| **Ours** | **0.144±0.012** | -0.003±0.036 |

independent manner, the plasticity when learning new systems is guaranteed. Second, the independency of the subnetworks also eliminates the forgetting issue and protect the performance stability on previously learned systems.

### 4.6 IN-DEPTH INVESTIGATION ON THE LEARNING DYNAMICS (RQ3,4)

Table 1 and 2 provide the overall performance. However, as introduced in Section 4.2, to obtain an in-depth understanding of the performance of different methods, we have to seek help from the performance matrix. In Figure 6, we visualize the performance matrices of different methods. The $i$-th column demonstrates the performance of the $i$-th task when learning sequentially over the systems. Comparing MS-GODE ((a) and (e)) and the other methods, we find that MS-GODE maintains a much more stable performance. EWC ((c) and (f)) also maintains a relatively stable performance of each system based on its regularization strategy. However, compared to MS-GODE, the model becomes less and less adaptive to new systems, because the regularization is applied to more parameters when proceeding to each new task. Fine-tune ((b) and (g)) is not limited by regularization, therefore is more adaptive on new systems but less capable of preserving the performance on previous systems compared to EWC. LwF (d), although based on knowledge distillation, does not directly limit the adaptation of the parameters. Finally, based on memory and gradient modification, GEM (h) maintains the performance better than fine-tune (g), and is more adaptive to new tasks than EWC (f). More details on the performance matrices are provided in Appendix A.5.

### 5 CONCLUSION

In this paper, we systematically study the problem of continual dynamics learning (CDL) from different perspectives, including investigating the influence of task configuration and evaluating the performance of existing continual learning techniques. Based on the findings, we propose Mode-switching Graph ODE (MS-GODE). Additionally, we also construct Bio-CDL, consisting of biological cellular systems and significantly enriching the research field of machine learning on system dynamics. Finally, we conduct comprehensive experiments on both physics and cellular system sequences, which not only demonstrate the effectiveness of MS-GODE, but also provide insights into the problem of machine learning over system sequences with dynamics shift.

ACKNOWLEDGMENTS

This project is supported by the National Research Foundation, Singapore, under its NRF Professorship Award No. NRF-P2024-001.

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

## A    APPENDIX / SUPPLEMENTAL MATERIAL

### A.1    SYSTEM SEQUENCE CONFIGURATION

#### A.1.1    PHYSICS SYSTEM SEQUENCE

Simulated physics systems are commonly adopted to evaluate the machine learning models in the task of learning system dynamics (Liu et al., 2024; Battaglia et al., 2016; Huang et al., 2020). The physics systems adopted in this work include spring connected particles and charged particles (Figure 7) with disparate dynamics, therefore are ideal for constructing system sequences to evaluate a model's continual learning capability under severe significant dynamics shift. Besides, the configuration of each system type is also adjustable. For the spring connected particles, the number of particles, strength of the springs, and the size of the box containing the particles are adjustable. For the charged particles, the number of particles, charge sign, and the size of box are adjustable. In our experiments, we constructed multiple systems with different configurations, which are aligned into sequences for the model to learn.

The physics system sequences are constructed to have different types of dynamics changes. System sequence 1 is composed of 8 spring connected particle systems. Each system contains 5 particles, and some pairs of particles are connected by springs. An illustration is given in Figure 7. For each system, besides the number of particles, the size of the box containing the particles and the strength of spring are adjustable. In Sequence 1, the first 4 systems have constant spring strength and decreasing box size. From the 5-th system, the box size is fixed, and the spring strength is gradually increased. Sequence 2 also contains 8 systems of spring connected particles and is designed to posses more severe dynamics shift. Specifically, both the box size and spring strength vary from the first to the last system, and the values are randomly aligned instead of monotonically increasing or decreasing. Sequence 3 is designed to posses more significant dynamics shift than Sequence 2 by incorporating the charged particle systems in the sequence. Specifically, Sequence 3 contains 4 spring connected particle systems and 4 charged particle systems, which are aligned alternatively. The box size gradually decreases and the interaction strength (spring strength or amount of charge on the particles) gradually increases. In a charged particle system, the particles could carry either positive or negative charge, and the system dynamics is governed by electrostatic force, which is significantly different from the spring connected particle system.

| System | S1 | S2 | S3 | S4 | S5 | S6 | S7 | S8 | S9 | S10 |
|---|---|---|---|---|---|---|---|---|---|---|
| Type | Spring | Spring | Spring | Spring | Spring | Spring | Spring | Spring | Spring | Spring |
| # particles | 5 | 5 | 5 | 5 | 5 | 5 | 5 | 5 | 5 | 5 |
| Box size | 10.0 | 5.0 | 3.0 | 1.0 | 0.5 | 0.5 | 0.5 | 0.5 | 3.0 | 1.0 |
| Interaction strength | 0.01 | 0.01 | 0.01 | 0.01 | 0.01 | 0.1 | 0.5 | 1.0 | 0.1 | 0.5 |

Table 3: Spring connected system configurations.

| System | C1 | C2 | C3 | C4 |
|---|---|---|---|---|
| Type | Charge | Charge | Charge | Charge |
| # particles | 5 | 5 | 5 | 5 |
| Box size | 10.0 | 3.0 | 1.0 | 0.5 |
| Interaction strength | 0.01 | 0.1 | 0.5 | 1.0 |

Table 4: Charged particle system configurations.

Specifically, we list the configurations of the systems in Table 3 and 4. Then the three sequences can be precisely represented as:

1. Sequence 1: $S_1 \to S_2 \to S_3 \to S_4 \to S_5 \to S_6 \to S_7 \to S_8$
2. Sequence 2: $S_1 \to S_8 \to S_2 \to S_7 \to S_3 \to S_6 \to S_5 \to S_5$
3. Sequence 3: $S_1 \to C_1 \to S_9 \to C_2 \to S_{10} \to C_3 \to S_8 \to C_4$

For each system, the simulation runs for 6,000 steps, and the observation is sampled every 100 steps, resulting in a 60-step series. During training, the first 60% part of the trajectory of each system is fed to the model to generate prediction for the remaining 40%. For each system sequence, 1,000 sequences are used for training, and another 1,000 sequences are used for testing.

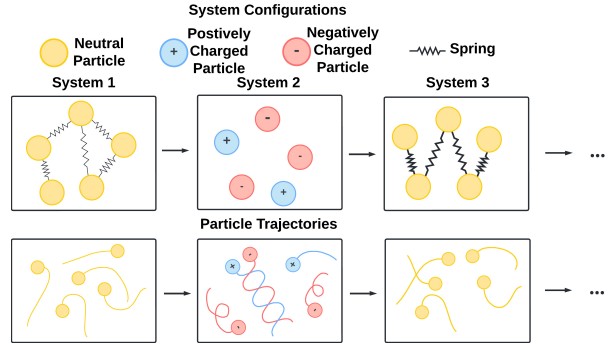

Figure 7: Illustration of the continual learning over different physics systems with different dynamics. The factors determining the dynamics shown in this figure include the type and strength of the interactions.

### A.1.2 BIOLOGICAL CELLULAR SYSTEM

In this paper, we build a novel benchmark, Bio-CDL, containing biological cellular dynamic systems based on Virtual Cell (Schaff et al., 1997; Cowan et al., 2012; Blinov et al., 2017) with different system configurations and variable selection. Currently, Bio-CDL is built based on two types of cellular models. The first one is rule-based model of EGFR receptor interaction with two adapter proteins Grb2 and Shc. The second is a compartmental rule based model of translocation through the nuclear pore of a cargo protein based on the Ran protein (GTPase). An illustration of the Ran system is provided in Figure 8.

Detailed description of these two systems can be found via `https://vcell.org/webstart/VCell_Tutorials/VCell6.1_Rule-Based_Tutorial.pdf` and `https://vcell.org/webstart/VCell_Tutorials/VCell6.1_Rule-Based_Ran_Transport_Tutorial.pdf`. Based on these 2 types of models, we construct multiple systems, which are aligned into different system sequences. Details are provided below.

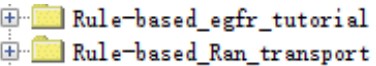

Figure 9

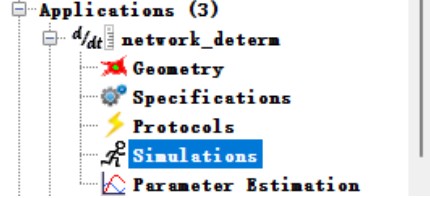

Figure 10

Our new benchmark is constructed based on the EGFR receptor interaction model and Ran transportation model. The length of the observe trajectories ranges from 250 to 550 time steps for EGFR systems and 40 to 150 time steps for Ran systems.

We provide below the details of data generation, and will release our generated data later. The code for generating the configuration files for the simulations is contained in

```
VCell_config_gen.py.
```

We have restricted the model coefficients to a specific range to ensure the simulations remain stable. After generating the configuration files, they should be uploaded into the VCell platform for generating the simulations. The specific steps are listed below.

1. Download the VCell client through https://vcell.org/run-vcell-software, create a free account and log in. 2. Select the 'BioModel' tab, in the 'Search' box, locate and load the models 'Rule-based_egfr_tutorial' or 'rule-based_Ran_transport' under the 'Tutorials' directory for the EGFR model or Ran model, respectively (Figure 9).

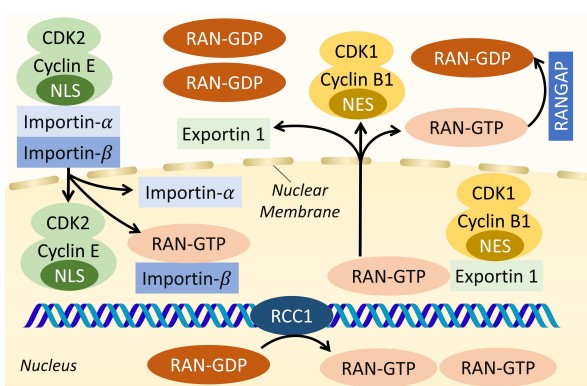

Figure 8: Illustration of the RAN-regulated nucleocytoplasmic transport (Moore, 2013). Briefly speaking, this model depicts the translocation of cargo proteins (*Exportin 1*) via nuclear pores with the assistance of *RAN* proteins. *RAN-GDP* is first activated into *RAN-GTP*) and then binds to cargo molecules (Exportin 1) forming a complex containing *RAN-GTP* and *Exportin 1*. Next, the complex is translocated across the *nuclear membrane* into the cytoplasm with the assistance of *RAN*. Finally, *RAN* and *Exportin 1* are dissociated after the translocation.

4. In the upper left part of the window, click the 'Applications', then click the 'Simulations' under the 'network_determ' tab (Figure 10).

4. In the main window on the upper right, select the 'Simulations' tab and click the first icon under the 'Simulations' tab to create a new simulation (Figure 11).

5. Click to select the created simulation, then click the icon with a blue arrow and a gear to load the previously generated simulations. 6. After the simulations are done, select all the simulations and click the icon with a green arrow and a gear to export all the simulation results into a specified folder. 7. Specify the

```
data_store_path
```

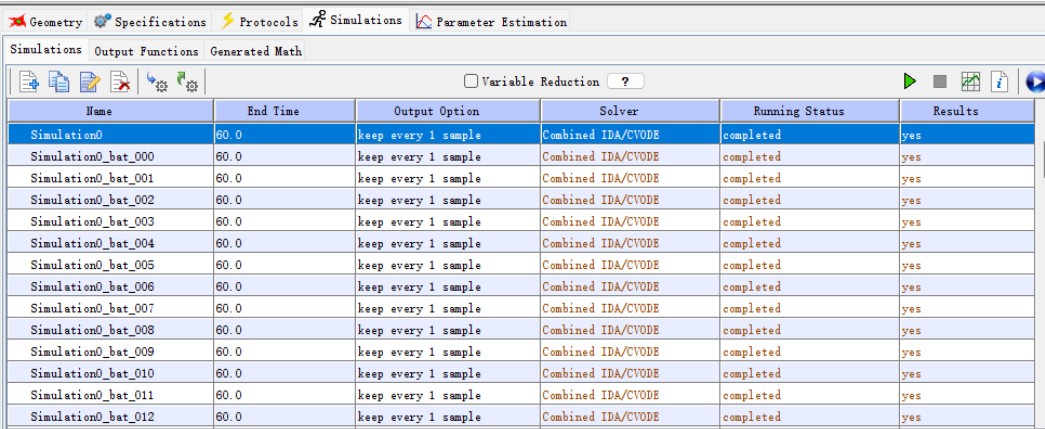

Figure 11

and run

```
generate_dataset.py
```

to obtain the data files that can be loaded and used by the MS-GODE model.

This implementation of MS-GODE is based on [Pytorch Geometric]`https://github.com/rusty1s/pytorch_geometric` API.

Example command for running the experiments with MS-GODE:

```
python run_models.py --cut_num 20 \
--nepos 20 \
--device 5 \
--n_iters_to_viz 10 \
--thresholding 'fast' \
--mask True \
--dropout_mask '0.0' \
--batch-size 10 \
--mode "ex" \
--normalizeVCellfeat 'universal' \
--system 'VCell' \
--repeats 5 \
--overwrite_results 'False' \
--fix_random_seed 'False' \
--save_results True
```

Example scripts for data generation of cellular/physics systems:

```
python ./data/generate_dataset.py --simulation VCell \
  --num-train 3000 \
  --num-test 3000 \
  --n-balls 5
```

The generation of cellular simulation data requires first obtaining simulation data from VCell platform, which is introduced in the Appendix A.1.2 of the submission.

```
python ./data/generate_dataset.py --simulation simulation \
  --num-train 3000 \
  --num-test 3000 \
  --n-balls 5
```

Setup

The mdoel implementation is based on the following packages:

- [Python 3.6.10](https://www.python.org/)

- [Pytorch 1.4.0](https://pytorch.org/)

- [pytorch_geometric 1.4.3](https://pytorch-geometric.readthedocs.io/)

- torch-cluster==1.5.3 - torch-scatter==2.0.4 - torch-sparse==0.6.1

- [torchdiffeq](https://github.com/rtqichen/torchdiffeq)

- [numpy 1.16.1](https://numpy.org/)

In our experiments, we adjust the parameter configurations of these two types of models to construct multiple systems with different dynamics, which are aligned into a sequence for the experiments. We generate simulated data using the Virtual Cell platform (VCell) and pre-process the data. In experiments, we adopt a 4-system sequence using the simulated data by alternatively align 2 EGFR systems and 2 Ran systems. EGFR system is rule-based model of EGFR receptor interaction with two adapter proteins Grb2 and Shc. Ran system is a compartmental rule based model of translocation through the nuclear pore of a cargo protein based on the Ran protein (GTPase). For constructing systems with different dynamics, different EGFR and Ran systems are constructed by selecting a subset of the observed variables from the complete systems. This not only enrich the dynamics diversity across different systems in the learning sequence, but also enhance the learning difficulty over the systems. The IDs (inherited from VCell platform) of the selected variables in $EGFR_1$, $EGFR_2$, $Ran_1$, $Ran_2$ are {s15,s16,s21,s22,s29,s33,s34, s38, s45, s47, s59, s74, s9, s96}, {s24, s25, s26, s27, s30, s36, s43, s44, s58, s68}, {Ran_C_nuc, RCC1, s10, s11, s12, s13, s14, s15, s16, s17, s18, s19, s2, s20, s21, s22, s23, s24, s25, s26, s27, s28, s29, s3, s30, s31, s32, s33, s34, s35, s4, s5, s6, s7, s8, s9 }, {s14, s15, s16, s23, s24, s25, s31, s5 }, respectively.

Using the code for generating cellular system data requires the following steps.

1. Using the script contained in our provided code 'MS-GODE/VCell_config_gen' to generate the configuration files for VCell simulation. Users can freely adjust the configurations in the script.

2. Download the VCell application from `https://vcell.org/`.

3. Load the generated configuration files into the VCell platform.

4. Generate simulation results. Instruction of using this script is also provided in the ReadMe file of our code.

5. Using the script 'MS-GODE/data/generate_dataset.py' for processing the simulation results. After this step, the data can be used for experiments.

Since the entire dataset is much larger than the 100 MB limit, we cannot provide the data in the supplementary materials. But we have provided all code and instruction for generating the data, and will release the complete Bio-CDL benchmark to the public later. Besides the sequence $EGFR_1 \rightarrow Ran_1 \rightarrow EGFR_2 \rightarrow Ran_2$ adopted in the experiments, the complete Bio-CDL also contain more different system configurations and sequence configurations. Moreover, we are also working on enriching Bio-CDL with more diverse system types and system sequences.

For each cellular system, we generate 100 iterations of simulation using VCell. Different from the physics systems, the temporal interval between two time steps of cellular system is not constant. Similar to the physics systems, the first 60% part of the observation of each system simulation is fed into the model to reconstruct the remaining 40% during training. For each system, we use 20 trajectories for training and 20 trajectories for testing.

### A.1.3 ADDITIONAL DETAILS OF EXPERIMENTAL SETTINGS

For the encoder network, the number of layers is 2, and the hidden dimension is 64. 1 head is used for the attention mechanism. The interaction network of the generator is configured as 1-layer network, and the number of hidden dimensions is 128. Finally, the decoder network is a fully-connected

layer. The model is trained for 20 epochs over each system in the given sequence. We adopt the AdamW optimizer (Loshchilov & Hutter, 2017) and set the learning rate as 0.0005. In our work, task boundaries are provided to the models during training.

### A.1.4 ADDITIONAL DISCUSSION ON PERFORMANCE AND LOW-LEVEL DYNAMICS SHIFT

The performance of MS-GODE is mainly determined on two factors, including the performance on each system and the forgetting issue. Therefore, to further improve the performance under gradual dynamics shift, there are two promising approaches:

1. Enhancing the performance on each single system. As revealed in Ramanujan et al. (2020), the larger the backbone network, the more probable the masked sub-network can reach the capacity of the full backbone network. In other words, increasing the size of the model can improve the performance.

2. Reducing the performance decrease after learning new systems. Since MS-GODE completely separate the masks for different dynamics, the forgetting issue has been eliminated. Therefore, the performance decrease after learning new systems mainly comes from the incorrect mask selection. The strategy above to increase model size can help improve the mask selection, because better fitting to each system increases the difference between the masks. Furthermore, the mask selection can also be improved by incorporating a mixture of selection criteria. Specifically, in our experiments, the mode-switching module of MS-GODE selects the mask based on the error of reconstructing the second 50% of the observation. This can be enhanced by incorporating different splitting ratios and mixing the result. Since the correct mask tend to exhibit lower error with most splitting ratios, selecting the one that succeeds in more splitting ratios could increase the possibility of finding the correct one.

### A.2 EDGE-POPUP ALGORITHM

In this work, we adopt the edge-popup algorithm (Ramanujan et al., 2020) for optimizing the binary masks. The main idea is to optimize a continuous score value for each entry of the mask during the backpropagation, and binarize the values into discrete binary values during forward propagation (Appendix A.2). Accordingly, the strategy for binarizing the mask entry values is a crucial factor influencing the performance. For convenience of the readers, we provide the details about this algorithm in this subsection.

Given a fully connected layer, the input to a neuron $v$ in the $l$-th layer can be formulated as a weighted summation of the output of the neurons in the previous later,

$$I_v = \sum_{u \mathcal{V}^{l-1}} w_{uv} z_u, \tag{16}$$

where $\mathcal{V}^{l-1}$ denotes the nodes in the previous layer and $z_u$ refers to the output of neuron $u$.

With the edge-popup strategy, the output is reformulated as

$$I_v = \sum_{u \mathcal{V}^{l-1}} w_{uv} z_u h(s_{uv}), \tag{17}$$

where $h(s_{uv})$ is the binary value over the weight $w_{uv}$ denoting whether this weight is selected and $s_{uv}$ the continuous score used in gradient descent based optimization. During backpropagation, the gradient will ignore $h(\cdot)$ and goes through it, therefore the gradient over $s_{uv}$ is

$$g_{s_{uv}} = \frac{\partial \mathcal{L}}{\partial I_v} \frac{\partial I_v}{\partial s_{uv}} = \frac{\partial \mathcal{L}}{\partial I_v} w_{uv} z_u, \tag{18}$$

where $\mathcal{L}$ denotes the loss function.

During forward propagation, $h(\cdot)$ can take different options as we mentioned and studied in Section 4.3. In our experiments 4.3, we tested both 'fast selection' and 'top-k selection' with different thresholds. 'Fast selection' set all entries with positive values into '1's and the other entries into '0's. 'Top-k selection' will rank the entry values and set a specified ratio of entries with the largest values into '1's.

### A.3 BASELINES

1. **Fine-tune** denotes using the backbone model without any continual learning technique.

2. **Elastic Weight Consolidation (EWC) (Kirkpatrick et al., 2017)** applies a quadratic penalty over the parameters of a model based on their importance to maintain its performance on previous tasks.

3. **Gradient Episodic Memory (GEM) (Lopez-Paz & Ranzato, 2017)** selects and stores representative data in an episodic memory buffer. During training, GEM will modify the gradients calculated based on the current task with the gradient calculated based on the stored data to avoid updating the model into a direction that is detrimental to the performance on previous tasks.

4. **Learning without Forgetting (LwF) (Li & Hoiem, 2017)** is a knowledge distillation based method, which minimizes the discrepancy between the the old model output and the new model output to preserve the knowledge learned from the old tasks.

5. **Bias Correction based Memory Replay (BCMR) (Chrysakis & Moens, 2023)**. This baseline is constructed by integrating the navie memory replay with the data sampling bias correction strategy (Chrysakis & Moens, 2023). In other words, the method does not train the data immediately after observing the data. Instead, it stores the observed data into a memory buffer. Whenever testing is required, the model will be trained over all buffered data.

6. **Scheduled Data Prior (SDP) (Koh et al., 2023)** considers that the importance of new and old data is dependent on the specific characteristics of the given data, therefore balance the contribution of new and old data based on a data-driven approach. Since the condition assumed for $c^2$ in SDP (Koh et al., 2023) does not align with the actual situation in our experiments, the parameter $\alpha$ and $\beta$ is Eq. (9) of SDP) (Koh et al., 2023) are manually selected in our experiments.

7. **Joint Training (Joint)** jointly trains a given model on all data instead of following the sequential continual learning setting.

### A.4 MODEL EVALUATION

Different from standard learning setting with only one task to learn and evaluate, in our setting, the model will continually learn on a sequence of systems, therefore the setting and evaluation are significantly different. In the model training stage, the model is trained over a system sequence. In the testing stage, the model will be tested on all learned tasks. Therefore the model will have multiple performance corresponding to different tasks, and the most thorough evaluation metric is the performance matrix $\mathrm{M}^p \in \mathbb{R}^{N \times N}$, where $\mathrm{M}^p_{i,j}$ denotes the performance on the $j$-system after learning from the 1-st to the $i$-th system, and $N$ is the number of systems in the sequence. In our experiments, each entry of $\mathrm{M}^p$ is a mean square error (MSE) value. To evaluate the overall performance on a sequence, the average performance (AP) over all learnt tasks after learning multiple tasks could be calculated. For example, *i.e.*, $\frac{\sum_{j=1}^{i} \mathrm{M}^p_{N,j}}{N}$ corresponds to the average model performance after learning the entire sequence with $N$ tasks. Similarly, the average forgetting (AF) after $N$ tasks can be formulated as $\frac{\sum_{j=1}^{N-1} \mathrm{M}^p_{N,j} - \mathrm{M}^p_{j,j}}{N-1}$. These metrics are widely adopted in continual learning works (Chaudhry et al., 2018; Lopez-Paz & Ranzato, 2017; Liu et al., 2021; Zhang et al., 2023a; Zhou & Cao, 2021), although the names are different in different works.

For convenience, the performance matrix can be visualized as a color map (Figure 6). For example, they are named as Average Accuracy (ACC) and Backwarde Transfer (BWT) in (Chaudhry et al., 2018; Lopez-Paz & Ranzato, 2017), Average Performance (AP) and Average Forgetting (AF) in (Liu et al., 2021), Accuracy Mean (AM) and Forgetting Mean (FM) in (Zhang et al., 2023a), and performance mean (PM) and forgetting mean (FM) in (Zhou & Cao, 2021).

### A.5 PERFORMANCE MATRIX ANALYSIS

Given a visualized performance matrix, we should approach it from two different dimensions. First, the $i$-th row of the matrix denotes the performance on each previously learned system after the

model has learned from the 1-st system to the $i$-th system. Second, to check the performance of a specific system over the entire learning process, we check the corresponding column. For example, in Figure 6 (e), each column maintains a stable color from the top to the bottom. This indicates that the performance of each system is perfectly maintained with little forgetting. But if the color becomes darker and darker from top to bottom (increasing MSE), it indicates that the corresponding method is exhibiting obvious forgetting problem.

