# OpenReview forum: "Learning system dynamics without forgetting"
_ICLR.cc/2025/Conference — ICLR 2025 Poster_

### Official Review · Reviewer_Cscj · 2024-10-30

**Soundness:** 3
**Presentation:** 3
**Contribution:** 3
**Rating:** 8
**Confidence:** 3

**Summary:**

The paper introduces Mode-switching Graph Ordinary Differential Equations (MS-GODE), a model designed to address Continual Dynamics Learning (CDL) for predicting time-dependent systems with evolving dynamics. Traditional approaches struggle with continual learning when dynamics vary, leading to issues like catastrophic forgetting, where previous dynamics are “forgotten” once new patterns are introduced. In this work, MS-GODE incorporates sub-network learning with mode-switching capabilities to prevent forgetting, allowing the model to dynamically adapt to different systems without overwriting learned dynamics. Additionally, the paper presents Bio-CDL, a benchmark dataset focusing on biological systems with varied dynamics, to further assess CDL models.

**Strengths:**

- The methodological foundation of MS-GODE is robust, integrating sub-network learning and mode-switching mechanisms to handle evolving system dynamics. The model architecture is well-constructed, and the choice to use fixed backbone weights with adaptive binary masks for each dynamic mode is theoretically sound and practically effective.
- Besides, The paper is generally well-organized, with clear problem framing and concise descriptions of the core components of MS-GODE.

**Weaknesses:**

- The model’s reliance on multiple components—such as sub-networks, binary masks, and a mode-switching module—adds to its complexity. While effective, this multi-faceted design may limit accessibility, especially for researchers less familiar with CDL or advanced dynamic modeling techniques.
- Although the Bio-CDL benchmark dataset represents a practical biological context, additional real-world datasets across broader domains (e.g., climate science, economics) could have bolstered the model’s credibility and practical relevance. The biological focus, while useful, limits insights into how the model would handle other types of complex, evolving systems.
- The MS-GODE model’s performance is sensitive to the initialization of binary masks and dropout rates, as shown in the experiments.
- The content in the main body about the benchmark introduced in this work is too short. As it is claimed to be a main contribution of this work, it would be better to describe more on the benchmark.

**Questions:**

1. How robust is the model to variations in initial mask configurations? Would pre-training the masks on related dynamics (e.g., similar system behaviors) improve or stabilize performance?
2. With multiple modes and sub-networks, how does the model handle scaling to larger or more complex networks (e.g., with thousands of nodes)? Would the approach require significant modification for larger systems, or is it inherently scalable?
3. The paper notes that abrupt dynamics shifts yield better performance than gradual shifts. Could the authors clarify whether any architectural modifications could improve MS-GODE’s performance under gradual shifts, which are common in many real-world applications?
4. What if some interactions require longer time to have an effect? Will the method still work?
5. What if the switching of the dynamics is not observed in the trajectories? Will the performance of the proposed method drop?
6. Minor: Please check the citation form in the paper, especially in introduction. They are not properly cited and break the sentences.

**Details Of Ethics Concerns:**

N.A.

---

> ### Author Response · Authors · 2024-11-20
> **Responses to Reviewer Cscj (Part 1)**
>
> We sincerely thank the reviewer for appreciating our proposed method and the paper presentation. We have provided detailed responses to each concern of the reviewer. All revisions to the paper are colored in blue for clarity.
>
> **Q1. The model’s reliance on multiple components—such as sub-networks, binary masks, and a mode-switching module—adds to its complexity. While effective, this multi-faceted design may limit accessibility, especially for researchers less familiar with CDL or advanced dynamic modeling techniques.**
>
> **A1.** Thanks for this nice concern on the accessibility of our work to researchers unfamiliar with this field.
>
> In our work, we have actually made efforts to boost the accessibility from the following perspectives.
>
> 1. In both main paper and Appendices, we have provided detailed introductions on our proposed MS-GODE from model design to hyperparameter configuration.
>
> 2. Researchers unfamiliar with the context can also easily deploy our model with the detailed instructions in our GitHub repository (link to the repository is provided in the general response above, which is visible to reviewers and ACs only).
>
> 3. To further facilitate the deployment of MS-GODE with researchers' own data, we have also updated the instructions in our GitHub repository with an additional section 'Using custom data'. Researchers can apply MS-GODE on their own data by simply reformatting the data following the instructions.
>
> 4. We will keep maintaining the repository to assist researchers in using our model and benchmark.
>
> Above all, for mature researchers, we have provided detailed instructions from both methodological and technical perspectives. For researchers from other domains, we have also provide easy-to-access instructions for them to customize the model towards their domains.
>
>
> **Q2. Although the Bio-CDL benchmark dataset represents a practical biological context, additional real-world datasets across broader domains (e.g., climate science, economics) could have bolstered the model’s credibility and practical relevance. The biological focus, while useful, limits insights into how the model would handle other types of complex, evolving systems.**
>
> **A2.** Thanks for this nice suggestions.
> First, we would like to clarify that our experiments are not only conducted on Bio-CDL, but also on **two types physics systems** with both attractive (spring and charged particles with same sign) and repulsive (charged particles with different sign) force interactions (introduced in Section 4.1), which are widely adopted by existing works [1, 2]. We have also updated the Introduction (the last paragraph) to highlight the use of physics system in our work. The superior performance across all the different systems demonstrates that our model can successfully handle various types of dynamics. With experiments on data from two disparate domains (physics and biology), the dynamics studied in our experiments is as diverse as the state-of-the-arts works, which typically also adopt two data domains. For example [1] adopts physics and motion datasets, while [2] adopts physics and molecule dynamics datasets.
>
> However, we totally agree that including more different systems can further justify the value of our model. Incorporating the datasets from areas like climate science and economics in CDL study is non-trivial and requires significant efforts, because the data requires careful investigation to be properly split into different tasks for continual learning.
> We have been actively working on this, and will further enrich our paper with experiments on these diverse domains.
>
> [1] Kang, Beomseok, et al. "Online Relational Inference for Evolving Multi-agent Interacting Systems." arXiv preprint arXiv:2411.01442 (2024).
>
> [2] Luo, Xiao, et al. "PGODE: Towards High-quality System Dynamics Modeling." Forty-first International Conference on Machine Learning.
>
> **Q3. The MS-GODE model’s performance is sensitive to the initialization of binary masks and dropout rates, as shown in the experiments.**
>
> **A3.** Thanks for this concern. The sensitivity to mask binarization (Figure 4) strategy and dropout rate (Figure 5) actually does not affect the robustness of the model.
>
> First, as shown in Figure 4, although adopting top-k strategy results in high sensitivity to the hyperparameter selection, adopting the fast mask selection strategy ensures consistently superior performance across different system sequences.
> Second, as revealed in Section 4.3 and Figure 5, setting the dropout rate to zero ensures an optimal performance empirically. This configuration has been adopted for experiments on all types system sequences in our work, and its effectiveness has been demonstrated by the consistently superior performance of MS-GODE across all systems including both physics systems and biological systems.
>
> In a word, the sensitivity experiments actually guided us to find out a robust model configuration across different systems.

---

> ### Author Response · Authors · 2024-11-20
> **Responses to Reviewer Cscj (Part 2)**
>
> **Q4. The content in the main body about the benchmark introduced in this work is too short. As it is claimed to be a main contribution of this work, it would be better to describe more on the benchmark.**
>
> **A4.** Thanks for this nice suggestion. We have enriched the description on the benchmark in Section 4.1 of the revised version.
>
>
> **Q5. How robust is the model to variations in initial mask configurations? Would pre-training the masks on related dynamics (e.g., similar system behaviors) improve or stabilize performance?**
>
> **A5.**
> Thanks for this insightful question. In our experiments, we have been using random initialization (Section 4.2, Baselines \& model settings), which has been studied by previous works [1]. As revealed by [1], the masked network is sensitive to the choice of the random initialization, and some initialization strategies are preferred. In our work, adopting Kaiming Normal initialization has been demonstrated to bring robust performance across different system dynamics. Furthermore, as revealed in [1], pre-training is also beneficial to stabilize the performance. However, different from the standard learning setting within only one task, mask pre-training is not suitable for our work under the continual learning setting. On the one hand, in the context of continual learning, we assume that the model has no access to the information of the future tasks (systems). On the other hand, continual learning requires the model to learn over a sequence of tasks with different dynamics, pre-training over one type of dynamics will only benefit one system, which will biases the performance towards one type of system dynamics and confuses us when comparing the performance on different dynamics.
>
> [1] Ramanujan, Vivek, et al. "What's hidden in a randomly weighted neural network?." Proceedings of the IEEE/CVF conference on computer vision and pattern recognition. 2020.
>
> **Q6. With multiple modes and sub-networks, how does the model handle scaling to larger or more complex networks (e.g., with thousands of nodes)? Would the approach require significant modification for larger systems, or is it inherently scalable?**
>
> **A6.** Yes, it is inherently scalable to large and complex networks.
>
> First, the size of our model is independent of the size of the systems (number of nodes). In experiments, we use the same model with a fixed size across different physics and biological systems with different number of nodes (Section A.1.3).
>
> Second, the extra memory consumption of the modes and sub-networks is negligible. In our MS-GODE, each sub-network (or each mode) is a combination of the backbone model and a binary mask. The backbone model is shared across all sub-networks, therefore does not require extra memory consumption. While the masks only contain binary values, therefore the memory consumption is negligible.

---

> ### Author Response · Authors · 2024-11-20
> **Responses to Reviewer Cscj (Part 3)**
>
> **Q7. The paper notes that abrupt dynamics shifts yield better performance than gradual shifts. Could the authors clarify whether any architectural modifications could improve MS-GODE’s performance under gradual shifts, which are common in many real-world applications?**
>
> **A7.** This is a very insightful question.
>
> First of all, we would like to clarify that the phenomenon of performance decrease on system sequence with low-level dynamics shift (gradual shift) is only significant in certain baselines. While for MS-GODE, although this phenomenon is observable, its influence is actually negligible. Specifically, As shown in Table 1, the performance of MS-GODE on the sequence with low-level dynamics shift (0.200) is already very close to Joint (0.194), which is the performance upper bound. In other words, although this phenomenon is broadly observed in the performance of different methods, including MS-GODE, it is not significant for MS-GODE and the performance of MS-GODE is still reliable.
>
> However, it is also promising to further enhance the performance.
> The performance of MS-GODE is mainly determined on two factors, including the performance on each system and the performance decrease after learning new systems. Accordingly, to further improve the performance under gradual dynamics shift, there are two potential approaches:
>
> 1. *Enhancing the performance on each single system*. As revealed in [1], as the width of the backbone network increases, the probability that there exists a subnetwork with same performance as a normally trained backbone network grows quickly. In other words, increasing the width of the model can improve the performance.
>
> 2. *Reducing the performance decrease after learning new systems*. Since MS-GODE completely separate the masks for different dynamics, the forgetting issue has been eliminated. Therefore, the performance decrease after learning new systems mainly comes from the incorrect mask selection (Section 4.4). The strategy above to increase model size can help improve the mask selection, because better fitting to each system increases the difference between the masks. Furthermore, the mask selection can also be improved by incorporating a mixture of selection criteria. Specifically, in our experiments, the mode-switching module of MS-GODE selects the mask based on the error of reconstructing the second 50\% of the given observation. This can be enhanced by incorporating different splitting ratios and mixing the result. Since the correct mask tend to exhibit lower error with most splitting ratios, selecting the one that succeeds with more splitting ratios could increase the possibility of finding the correct one.
>
> Above all, although MS-GODE has already obtained superior performance on system sequence with gradual dynamics shift, there are also several strategies to further enhance its performance.
>
> We have created a new section to clarify this issue in the revised version (A.1.4).
>
> [1] Ramanujan, Vivek, et al. "What's hidden in a randomly weighted neural network?." Proceedings of the IEEE/CVF conference on computer vision and pattern recognition. 2020.
>
> **Q8. What if some interactions require longer time to have an effect? Will the method still work?**
>
> Yes, the method will definitely work.
> As introduced in abstract and introduction section, we focus on observation-based trajectory prediction. If some interactions currently have no effect on the trajectories, they will not be captured by MS-GODE or any observation-based prediction model. Whenever they start to influence the trajectories of the system, the dynamics of the system is changed (mode-switching), and our MS-GODE will encode the new dynamics with a new mask.
>
>
>
>
> **Q9. What if the switching of the dynamics is not observed in the trajectories? Will the performance of the proposed method drop?**
>
> **A9.**
> The performance will not drop.
> Dynamics refers to the underlying rules that determine how the trajectories evolve, i.e. how previous trajectories lead to future trajectories. Therefore, switching of dynamics equals that the trajectories start to evolve in a different way. In other words, switching of dynamics is same as change in trajectories, therefore will definitely be observed in trajectories.
>
> Moreover, not observing any change in the trajectories means that the model can always predict the future trajectories in the same way, and the performance will not be influenced at all.
>
>
>
> **Q10. Minor: Please check the citation form in the paper, especially in introduction. They are not properly cited and break the sentences.**
>
> **A10.** Thanks for reminding us about this issue. We have carefully checked and corrected all the citation forms in the revised paper.

---

> > ### Comment · Reviewer_Cscj · 2024-11-22
> >
> > Dear Authors,
> >
> > I appreciate the rebuttal and the revised paper. My concerns are addressed with pretty many details. I raise the score to 8 to support the acceptance of this paper.
> >
> > Best regards.

---

> ### Author Response · Authors · 2024-11-22
> **Thank you for raising the score to support our work!**
>
> Dear Reviewer,
>
> We are delighted to hear that your concerns have been addressed, and we sincerely appreciate your kind recognition of our work!
>
> Best regards,
>
> Authors

---

### Official Review · Reviewer_ZNCe · 2024-11-03

**Soundness:** 2
**Presentation:** 3
**Contribution:** 2
**Rating:** 6
**Confidence:** 4

**Summary:**

This paper tackles a significant gap in dynamics learning by introducing Continual Dynamics Learning (CDL), addressing systems whose dynamics evolve over time. The core contribution, Mode-switching Graph ODE (MS-GODE), combines sub-network learning with mode-switching to handle varying dynamics while preventing catastrophic forgetting. Their approach marks the first systematic treatment of this problem, supported by extensive empirical validation.

**Strengths:**

The paper's primary innovation lies in recognizing and formalizing the CDL problem, which has been overlooked despite its practical importance. The MS-GODE architecture effectively combines proven techniques (Graph ODEs, sub-network learning) with a novel mode-switching mechanism, demonstrating strong performance across different system configurations. The introduction of Bio-CDL, a benchmark featuring biological cellular systems, significantly enriches the field beyond traditional physics simulations.

**Weaknesses:**

- Several important baselines are notably absent from the comparison, including CG-ODE, GG-ODE, PG-ODE, and HOPE.

- The experimental validation would be more convincing if it included tests on human motion and molecular dynamics (MD17) datasets, which could reveal how the approach handles different types of dynamic patterns. The model shows sensitivity to hyperparameters, particularly dropout rates and mask selection strategies, but lacks clear guidelines for parameter selection in practical applications.

- The paper also needs stronger theoretical foundations - while the empirical results are promising, there's no formal analysis of why mode-switching works better than traditional continual learning approaches.

**Questions:**

Contrary to the author's claim, there have been pretrained models for dynamical systems lately. Can they be leveraged to improve MS-GODE's performance, especially in sub-network learning?

Seifner, Patrick, et al. "Foundational inference models for dynamical systems." arXiv preprint arXiv:2402.07594 (2024).

---

> ### Author Response · Authors · 2024-11-20
> **Responses to Reviewer ZNCe (Part 1)**
>
> We sincerely thank the reviewer for recognizing our novel contribution in formalizing the CDL problem, the effectiveness of our proposed method, and our contribution in proposing a novel benchmark. We have provided detailed responses to each concern of the reviewer. All revisions to the paper are colored in blue for clarity.
>
>
> **Q1. Several important baselines are notably absent from the comparison, including CG-ODE, GG-ODE, PG-ODE, and HOPE.**
>
> **A1.**
> Thanks for recommending the four excellent works.
>
> We have added detailed discussions on these works in the revised version (Section 2). The discussion, as well as the comparison between these works and our MS-GODE, are also provided below for convenience. First, Coupled Graph ODE (CG-ODE) [1] applies ODE-based modeling to both node features and interaction between the system components, in order to capture and predict the evolution of both node features and the interactions. Generalized Graph Ordinary Differential Equations (GG-ODE) [2] focuses on systems with commonalities but exhibit different dynamics due to different environmental factors (exogenous factors). GG-ODE consists of two parts. One captures the commonalities to ensure generalizability while the other one captures the exogenous factors.
> Prototypical Graph ODE (PGODE) [3] disentangles object (system component) states and system states to independently model their influence on the system evolution. which improves the generalization capability of the model.
> High-order graph ODE (HOPE) [4] innovatively incorporates two types of high-order information, including information from high-order spatial neighborhood and high-order derivatives, into dynamical system modeling.
>
> Above all, these works and our MS-GODE focus on different perspectives of dynamical system learning. CG-ODE [1] and HOPE [4] focus on advanced frameworks for modeling single dynamical systems, while GG-ODE [2] and PGODE [3] focus on the generalization capability across systems whose dynamics factors can be disentangled and modeled separately. In contrast, MS-GODE learns over systems governed by different dynamics, with a focus on alleviating the forgetting issue over the previous systems.
>
> Moreover, since the code of CG-ODE [1] has been released, we are also able to compare CG-ODE with MS-GODE empirically.
>
> ||Seq1: Low-level dynamics shift||Seq2: Mid-level dynamics shift||Seq3: High-level dynamics shift||
> |-|-|-|-|-|-|-|
> ||AP $\downarrow$|AF $\downarrow$|AP $\downarrow$|AF $\downarrow$|AP $\downarrow$|AF $\downarrow$|
> |CG-ODE|0.353$\pm$0.019 | 0.120$\pm$0.013    | 0.361$\pm$0.029 | 0.319$\pm$0.021 	| 0.241$\pm$0.018 | 0.055$\pm$0.025|
> |LG-ODE|0.369$\pm$0.027 | 0.115$\pm$0.016    | 0.391$\pm$0.044 | 0.314$\pm$0.030 	| 0.258$\pm$0.025 | 0.086$\pm$0.037|
> |Joint|0.194$\pm$0.006 | -                             | 0.186$\pm$0.015 | -                          	 | 0.116$\pm$0.009 | -|
> |MS-GODE| 0.200$\pm$0.003 | 0.002$\pm$0.004 | 0.204$\pm$0.005 | -0.001$\pm$0.001 | 0.113$\pm$0.001 | -0.000$\pm$0.000|
>
> Table: Performance comparisons on physics system sequences ($\downarrow$ lower means better)
>
> [1] Huang, Zijie, Yizhou Sun, and Wei Wang. "Coupled graph ode for learning interacting system dynamics." Proceedings of the 27th ACM SIGKDD conference on knowledge discovery \& data mining. 2021.
>
> [2] Huang, Zijie, Yizhou Sun, and Wei Wang. "Generalizing graph ode for learning complex system dynamics across environments." Proceedings of the 29th ACM SIGKDD Conference on Knowledge Discovery and Data Mining. 2023.
>
> [3] Luo, Xiao, et al. "PGODE: Towards High-quality System Dynamics Modeling." Forty-first International Conference on Machine Learning.
>
> [4] Luo, Xiao, et al. "Hope: High-order graph ode for modeling interacting dynamics." International Conference on Machine Learning. PMLR, 2023.

---

> ### Author Response · Authors · 2024-11-20
> **Responses to Reviewer ZNCe (Part 2)**
>
> **Q2. The experimental validation would be more convincing if it included tests on human motion and molecular dynamics (MD17) datasets, which could reveal how the approach handles different types of dynamic patterns. The model shows sensitivity to hyperparameters, particularly dropout rates and mask selection strategies, but lacks clear guidelines for parameter selection in practical applications.**
>
> **A2.**
> Thanks for this suggestion, which is helpful to further improve our work. We have been actively conducting experiments over the CMU motion dataset [1] and MD17 [2] to further enrich our work.
>
> We would also like to highlight that our approach has **already been evaluated on various types of dynamic patterns**, and has exhibited superior performance and robustness. Specifically, we have conducted experiments on **two types physics systems** with both attractive (spring and charged particles with same sign) and repulsive (charged particles with different sign) force interactions, and **two types of biological systems** with disparate dynamics (Section 4.1 and Appendix A.1.2). The superior performance across all the different systems demonstrates that our model can successfully handle various types of dynamics patterns. The dynamics studied in our experiments is comparably or more diverse than existing state-of-the-arts works.
>
> **Sensitivity to hyperparameters:** Although the model is sensitive to dropout rates and mask selection strategies, configuring these hyperparameters does not require any effort as empirically revealed in our work. First, as shown in Figure 4, adopting the fast mask selection strategy does not require configuring any hyperparameter but ensures consistently superior performance across different system sequences. Second, as revealed in Section 4.3 and Figure 5, setting the dropout rate to zero ensures an optimal performance empirically. This configuration has been adopted for experiments on all types system sequences in our work, and its robustness has been demonstrated by the consistently superior performance of MS-GODE across all systems including both physics systems and biological systems.
>
>
> [1] Carnegie-mellon motion capture database. 2003. URL http://mocap.cs.cmu.edu.
>
> [2] Chmiela, Stefan, et al. "Machine learning of accurate energy-conserving molecular force fields." Science advances 3.5 (2017): e1603015.
>
> **Q3. The paper also needs stronger theoretical foundations - while the empirical results are promising, there's no formal analysis of why mode-switching works better than traditional continual learning approaches.**
>
>
> **A3.** Thanks for this suggestion. We have enriched our paper with formal analysis on the advantage of the proposed MS-GODE over the baselines (Section 4.5). This involves two crucial perspectives of continual learning, i.e. the stability-plasticity dilemma.
>
> 1. Plasticity. As analyzed in [1], as the width of the backbone network increases, the probability that there exists a subnetwork with same performance as a normally trained backbone network grows quickly. In other words, each system-specific subnetwork has the potential to match the plasticity of the full backbone network. In contrast, the baselines typically optimize the parameters of one single backbone network for dynamics of different systems. Accordingly, the plasticity of the network keeps decreasing as the number of learned systems increases. For example, EWC applies regularization on model parameters to protect the performance on learned systems, which limits the model's plasticity when learning new systems.
>
> 2. Stability. In MS-GODE, the system-specific subnetworks are completely independent, therefore the forgetting issue is completely avoided. In contrast, since the baselines typically optimize the same set of parameters for different systems, the learning over different system dynamics will always have certain level of interference with each other.
>
> In a word, MS-GODE achieves superior performance because it better resolves the stability-plasticity dilemma, which is the key challenge of continual learning.
>
> [1] Ramanujan, Vivek, et al. "What's hidden in a randomly weighted neural network?." Proceedings of the IEEE/CVF conference on computer vision and pattern recognition. 2020.

---

> ### Author Response · Authors · 2024-11-20
> **Responses to Reviewer ZNCe (Part 3)**
>
> **Q4. Contrary to the author's claim, there have been pretrained models for dynamical systems lately [1]. Can they be leveraged to improve MS-GODE's performance, especially in sub-network learning?**
>
> **A4.** Thanks for recommending this work. We have added discussion on [1] in our paper (Section 4.2). Although [1] provides valuable insights into dynamical system learning, it is currently inapplicable to our task.
>
> First, the settings adopted in [1] is different from ours. [1] studies the interpolation task, while our work focus on the extrapolation task. In other words, the pre-trained model in [1] is trained to recover missing data of the trajectories within the observed time period, while our task is to predict the future trajectories beyond the observed time period.
>
> Second, [1] is currently under review and the pre-trained model or the code has not been released yet.
>
> However, although the targeted tasks in [1] is different from ours, its pre-trained model has encoded knowledge from various dynamical systems. Therefore, we still believe it still has the potential to be leveraged for further improving our work. We will definitely investigate how to adapt the model for extrapolation tasks once it is released.
>
>
> [1] Seifner, Patrick, et al. "Foundational inference models for dynamical systems." arXiv preprint arXiv:2402.07594 (2024).

---

### Official Review · Reviewer_VBaq · 2024-11-06

**Soundness:** 2
**Presentation:** 3
**Contribution:** 2
**Rating:** 6
**Confidence:** 4

**Summary:**

The main motivation of this paper is that, system dynamics are not static, rather subject to change over time in practice. Accordingly, the authors developed a mode-switching graph ODE (MS-GODE) to continually learn evolving system dynamics. Unlike most of the existing works on dynamics modeling, which have been focused on a single dynamics, MS-GODE can process multiple system dynamics by automatically switching binary masks on shared model weights (i.e., different weights for different system dynamics). Also, the authors introduces bio-CDL, a novel dynamic system benchmark to evaluate the proposed methods in addition to physics-based particle systems.

**Strengths:**

- Continual learning setup on interacting system dynamics is a novel problem. Since very recent works have explored evolving system dynamics, continual learning can provide a new insight on the related research community.

- As the authors mentioned in L131, neural ODE has been focused on a single dynamical systems. Combination of neural ODE with masked networks is a new approach.

- The paper introduces a novel benchmark using biological cellular systems. Considering many benchmarks on interacting dynamical systems are based on physical systems (e.g., springs, charges), the new benchmark could provide another insight whether models still perform well when predicting dynamic states, not locations of particles.

- The details about the experimental settings and background are well presented.

**Weaknesses:**

- While I still consider the combination of neural ODE with continual learning interesting, technical novelties seem somewhat limited. For example, both the network design (e.g., LG-ODE, NRI) and learning method (e.g., masked-based CL, edge-popup algorithm) are already well known. I think, unique technical contributions this paper had made should have been clearly presented.

- Also, the motivation of CDL needs to be better clarified. I don’t think continual learning is always required for all evolving dynamical systems. For example, what if a system never repeats the past dynamics once it evolves? what if the prediction model can quickly enough adapt to new dynamics? We don’t need to make the models avoid catastrophic forgetting in these cases. In other words, unless the system dynamics repeat again and again, which might narrow the scope of target applications, mitigating catastrophic forgetting won't be useful.

- The results from Table 1 (performance of all methods is better in high-level dynamics shift) are interesting. The authors provide a brief discussion in L471-L475, but is there any experimental or theoretical evidence to believe this? For example, how do the authors believe that “e.g. Fine-tune, the results indicate that diverse systems may guide the model to exploit different parameters for different systems”? In other words, the discussion and detailed analysis on the performance is not enough while comparison with other methods in terms of AP and AF are well studied.

- There have been many recent works to address evolving system dynamics [1,2,3] though they are not continual learning-based works. Since this paper specifically address such systems, I think related works should have included them to further clarify the novelty in the problem setup.

[1] Dynamic Graph Neural Networks Under Spatio-Temporal Distribution Shift, NeurIPS 2022

[2] CARE: Modeling Interacting Dynamics Under Temporal Environmental Variation, NeurIPS 2023

[3] Online Relational Inference for Evolving Multi-agent Interacting Systems, NeurIPS 2024

- In equation (3), do the authors assume the spatial-temporal edges (or graph structures) are known?

- Can the authors specify if task boundaries should be known for training? As there are many different continual learning setups, it would be nice to specify it.

- There are some typos (e.g., L208: “to x) as a state” -> “to x as a state”, L965: “backpropagatioon” -> “backpropagation”)

- It would be better if a code implementation was provided in the review stage.

**Questions:**

provided with weakness

---

> ### Author Response · Authors · 2024-11-20
> **Responses to Reviewer VBaq (Part 1)**
>
> We sincerely thank the reviewer for recognizing the novelty of our target problem and our approach, as well as appreciating our contribution in benchmark and clarity in presentation. We have provided detailed responses to each concern of the reviewer. All revisions to the paper are colored in blue for clarity.
>
> **Q1. While I still consider the combination of neural ODE with continual learning interesting, technical novelties seem somewhat limited. For example, both the network design (e.g., LG-ODE, NRI) and learning method (e.g., masked-based CL, edge-popup algorithm) are already well known. I think, unique technical contributions this paper had made should have been clearly presented.**
>
> **A1.**
> Thanks for this nice suggestion on clarifying of our technical contribution.
>
> Our technical contribution consists of the following components.
> First, as recognized by the reviewer, we develop a novel strategy to integrate neural ODE and continual learning techniques for model training in the CDL scenario, which is empirically justified to be highly effective. Second, we extend the mask selection technique from classification tasks into regression tasks with the mode-switching module (Section 3.6).
>
> We have carefully improved the Introduction (revised parts are colored blue) of the paper to highlight our technical contribution, as well as our contribution in proposing the novel and practical continual dynamics learning (CDL) scenario, investigating the applicability of different techniques in CDL, and a novel benchmark for the dynamics learning community.
>
> **Q2. Also, the motivation of CDL needs to be better clarified. I don’t think continual learning is always required for all evolving dynamical systems. For example, what if a system never repeats the past dynamics once it evolves? what if the prediction model can quickly enough adapt to new dynamics? We don’t need to make the models avoid catastrophic forgetting in these cases. In other words, unless the system dynamics repeat again and again, which might narrow the scope of target applications, mitigating catastrophic forgetting won't be useful.**
>
> **A2.** Thanks for this suggestion on clarifying our motivation. We have enriched the motivation part in Introduction (the first paragraph) to clarify the issue on repeating and non-repeating dynamics.
>
> *What if a system never repeats the past dynamics once it evolves?* We agree that mitigating forgetting is unnecessary in systems that never repeat their dynamics, e.g. chaotic systems. However, we would like to clarify that many real-world systems exhibit repeated dynamics, and mitigating the forgetting is actually crucial in a broad range of scenarios. For example, dynamics of many physics systems are controlled by environmental factors, e.g. temperatures [1]. The values of these factors like temperatures typically oscillate within a certain range, therefore the dynamics of the systems will also repeat. This is also true in biological systems. Moreover, biological cellular systems also go through different phases of cell cycle [2], which can be captured by the different modes in our MS-GODE. Finally, the highly unstable stock market systems may also exhibit certain repeated patterns [3].
>
> *What if the prediction model can quickly enough adapt to new dynamics?* Both quick adaptation and reliable memory are two desired properties for continual learning models when facing new tasks. Even if a model is capable of quick adaption, repeated training over same system dynamics is inefficient at least in the following perspectives. First, when the dynamics is complex, significant amount of observational data may be required to faithfully reflect the underlying dynamics. Second, when the dynamics alters in a high frequency, repeated training the model frequently also incurs high computational cost. In this case, the efficiency will be significantly boosted if the previous knowledge can be directly leveraged. In a word, quick adaptation and reliable memory are not contradictory but actually complementary to each other. Both of them are crucial for continual learning models.
>
>
> [1] Huang, Zijie, Yizhou Sun, and Wei Wang. "Generalizing graph ode for learning complex system dynamics across environments." Proceedings of the 29th ACM SIGKDD Conference on Knowledge Discovery and Data Mining. 2023.
>
> [2] Norbury, Chris, and Paul Nurse. "Animal cell cycles and their control." Annual review of biochemistry 61.1 (1992): 441-468.
>
> [3] Dorr, Dietmar H., and Anne M. Denton. "Establishing relationships among patterns in stock market data." Data & Knowledge Engineering 68.3 (2009): 318-337.

---

> ### Author Response · Authors · 2024-11-20
> **Responses to Reviewer VBaq (Part 2)**
>
> **Q3. The results from Table 1 (performance of all methods is better in high-level dynamics shift) are interesting. The authors provide a brief discussion in L471-L475, but is there any experimental or theoretical evidence to believe this? For example, how do the authors believe that “e.g. Fine-tune, the results indicate that diverse systems may guide the model to exploit different parameters for different systems”? In other words, the discussion and detailed analysis on the performance is not enough while comparison with other methods in terms of AP and AF are well studied.**
>
> **A3.** Thanks for this nice concern. We have enriched the analysis on the performance and added discussion on the relationship between performance and levels of dynamics shift in the revised version (Section 4.4), which is also provided below.
>
> As shown in Table 1, when learning on the system sequence with high-level dynamics shift, lower AF (average forgetting) indicates less forgetting when learning on new systems. Two possible cases are: 1. The model is not well trained on any system (it has not encoded much information), therefore it has nothing to forget. 2. The model is well trained and can maintain enough information from each system. However, the lower AP (low MSE) obtained on the system sequence with high-level dynamics shift indicates that the model has well adapted to each system, i,e, case 1 is not true. When case 2 holds, it indicates that the model's parameters are sufficiently updated to fit each system. When learning on one system, two possible cases are: 1. All parameters are modified uniformly. 2. A subset of parameters are modified more than the others.
> When the baseline is Fine-tune, if all the parameters are uniformly modified, it would be almost impossible to maintain the performance on previous systems. Therefore, the potential case is that each system relies more on a subset of the parameters, and systems with larger difference in dynamics may lead to less overlap in the subsets of parameters they rely on.
>
> This phenomenon has actually been studied with both theoretical and empirical evidence in previous continual learning works [1,2]. For example, EWC [1] adopts the fisher information to evaluate the importance of the parameters to a learned task (i.e. system in our work), which means that different parameters are indeed encoding different amount of information of the learned task.
>
> [1] Kirkpatrick, James, et al. "Overcoming catastrophic forgetting in neural networks." Proceedings of the national academy of sciences 114.13 (2017): 3521-3526.
>
> [2] Aljundi, Rahaf, et al. "Memory aware synapses: Learning what (not) to forget." Proceedings of the European conference on computer vision (ECCV). 2018.
>
>
> **Q4. There have been many recent works to address evolving system dynamics [1,2,3] though they are not continual learning-based works. Since this paper specifically address such systems, I think related works should have included them to further clarify the novelty in the problem setup.**
>
> **A4.** Thanks for recommending the related works on addressing evolving system dynamics. We have added discussions on these works in the related works of the revised version (Section 2). The discussion is also provided below for convenience,
>
>
> Disentangled Intervention-based Dynamic graph Attention networks (DIDA) [1] disentangles the invariant and variant patterns in dynamic graphs, and leverages the invariant patterns to ensure a stable prediction performance under spatio-temporal distribution shift. Context-attended Graph ODE (CARE) [2] models the continuously varying environmental factors with a context variable, which is leveraged to better predict the system evolution with temporal environmental variation. Online Relational Inference (ORI) [3] models the relationship among the system components as trainable parameters, which is accompanied with a novel adaptive learning rate technique (AdaRelation) to ensure quick relational inference in the online setting.
>
> Above all, these three works and our MS-GODE focus on different perspectives of dynamical system learning. DIDA [1] and CARE [2] aim at disentangling the inherent invariant system dynamics and the environmental factors that cause the dynamics shift, thereby boosting the generalization of the prediction model across various environments. ORI [3] targets efficient identification of the interactions among the system components in the online setting, with a focus on fast adaption to new systems. In contrast, our MS-GODE learns over systems governed by different dynamics, with a focus on alleviating the forgetting issue over the previous systems.
>
>
> [1] Dynamic Graph Neural Networks Under Spatio-Temporal Distribution Shift, NeurIPS 2022
>
> [2] CARE: Modeling Interacting Dynamics Under Temporal Environmental Variation, NeurIPS 2023
>
> [3] Online Relational Inference for Evolving Multi-agent Interacting Systems, NeurIPS 2024

---

> ### Author Response · Authors · 2024-11-20
> **Responses to Reviewer VBaq (Part 3)**
>
> **Q5. In equation (3), do the authors assume the spatial-temporal edges (or graph structures) are known?**
>
> **A5.** Yes, the spatial-temporal edges are assumed to be known. As defined in Equation (2), a spatial-temporal edge is created between two states if their temporal distance is within a threshold and if their corresponding nodes are connected by spatial edges.
>
> **Q6. Can the authors specify if task boundaries should be known for training? As there are many different continual learning setups, it would be nice to specify it.**
>
> **A6.** Thanks for this suggestion. In our experiments, task boundaries are provided to the models during training. We have added discussion on the task boundaries in the revised paper (Section A.1.3).
>
>
> **Q7. There are some typos (e.g., L208: “to x) as a state” -> “to x as a state”, L965: “backpropagatioon” -> “backpropagation”)**
>
> **A7.** We have corrected these typos in the revised version, and also proofread the entire paper to eliminate other typos.
>
> **Q8. It would be better if a code implementation was provided in the review stage.**
>
> **A8.** Thanks for this suggestion. We have provided all code implementations for both our model and the baselines via the anonymous link in the general response above, which is visible to reviewers and ACs only.
>
> This repository also contains code and detailed instruction on using our novel benchmark.

---

> ### Author Response · Authors · 2024-11-24
> **A Friendly Reminder to Reviewer VBaq: Approaching Discussion Deadline**
>
> We sincerely thank the reviewer for the recognition of our work and providing valuable and constructive comments.
>
> We have made every effort to address all concerns raised, and have revised the paper wherever appropriate.
>
> With only three days remaining in the author-reviewer discussion period, we would greatly appreciate if the reviewer could let us know whether there are any remaining concerns.

---

> ### Author Response · Authors · 2024-11-29
> **We would really appreciate if the reviewer could let us know whether there are any remaining concerns**
>
> Dear reviewer,
>
> Again, we sincerely thank the reviewer for the recognition of our work and providing valuable and constructive comments to further improve our work.
>
> We have made every effort to address all concerns raised.
> Specifically, we have carefully revised our paper to clarify our contribution and motivation, added more analysis on the experimental results, added discussion on the recommended related works, clarified several details of our work, corrected several typos, and provided all code implementation of our work.
>
> With the approaching deadline of the author-reviewer discussion period, we would greatly appreciate if the reviewer could let us know whether there are any remaining concerns.
>
> Best regards,
>
> Authors

---

> ### Comment · Reviewer_VBaq · 2024-11-29
>
> Dear Authors,
>
> I apologize for the late response. I have carefully read the rebuttal and the revised manuscript. I appreciate the efforts from the authors to address my questions. I don't have significant concerns now so would like to raise the score to 6.
>
> Best regards,
>
> Reviewer VBaq

---

> > ### Author Response · Authors · 2024-11-29
> > **Thank you very much for supportting our work!**
> >
> > Dear reviewer,
> >
> > We are happy that all the concerns are resolved, and we sincerely thank you for raising the score and supporting our work!
> >
> > Best regards,
> >
> > Authors

---

### Official Review · Reviewer_Pzyn · 2024-11-11

**Soundness:** 3
**Presentation:** 3
**Contribution:** 3
**Rating:** 5
**Confidence:** 4

**Summary:**

The paper proposed a continual learning framework based on GraphODE over dynamical systems. The key motivation is to address the catastrophic forgetting problem if multiple systems of different configurations are within a sequence. The proposed Mode-switching Graph ODE (MS-GODE) is able to select the best sub-network mask for a given observation sequence, following the parameter-isolation category in continual learning. A new dataset on biological cellular systems is created in the experiment.

**Strengths:**

1. The paper studies an interesting problem in dynamical system modeling under distribution shifts. Though I feel the motivation example can be changed to more realistic one (see below).

2. The paper proposes a useful benchmark for evaluating the distribution shifts in dynamical system modeling, on biological cellular systems.

3. The proposed method is able to achieve good performance over selected baselines.

**Weaknesses:**

1. While I in general get the motivation of this paper, I feel the examples used in the introduction section need to be further improved. First for Figure 1, there are so many contents/fonts in the figure that are not well-explained in the caption or the main text. It is suggested to use for example legend to denote different kinetic factors, using some boxes to denote the overall system consists of multiple objects. Here for me, it is hard to read from this figure along with its current explanations. For the second example, I feel it is not very natural to combine spring systems and charged particle systems, as there is no reasonable transition among them. A more proper example may be that some water particles will froze into solid when the temperature decreases, and the interaction among liquid and solid particles can be very different. I encourage the authors to further improve the motivation example so the audience from various backgrounds can understand them easily.

2. For the baselines, the authors chose different continual learning framework as comparison. However for learning over multiple systems, there are some work [1,2] that directly learn a generalized neural simulators. It is suggested to also discuss their performance and show if continual learning is the better way to learning over different systems.

3. For majority of the citation, it should be \citep (reference with a brackets) instead of \cite. For example, line 40-50.

[1] Generalizing Graph ODE for Learning Complex System Dynamics across Environments.

[2] HOPE: High-order Graph ODE For Modeling Interacting Dynamics
[2]

**Questions:**

1. For the experiment table 1, what does the results fine-tune and joint mean respectively? In line 400-402 the authors mention that for each baseline there are two options, so I overall do not get the meaning of the two rows in Table 1,which model are they based on ?

---

> ### Author Response · Authors · 2024-11-20
> **Responses to Reviewer Pzyn (Part 1)**
>
> We sincerely thank the reviewer for appreciating the importance of our target problem, as well as recognizing our contribution in benchmark and performance improvement. We have provided detailed responses to each concern of the reviewer. All revisions to the paper are colored in blue for clarity.
>
> **Q1. While I in general get the motivation of this paper, I feel the examples used in the introduction section need to be further improved. First for Figure 1, there are so many contents/fonts in the figure that are not well-explained in the caption or the main text. It is suggested to use for example legend to denote different kinetic factors, using some boxes to denote the overall system consists of multiple objects. Here for me, it is hard to read from this figure along with its current explanations. For the second example, I feel it is not very natural to combine spring systems and charged particle systems, as there is no reasonable transition among them. A more proper example may be that some water particles will froze into solid when the temperature decreases, and the interaction among liquid and solid particles can be very different. I encourage the authors to further improve the motivation example so the audience from various backgrounds can understand them easily.**
>
> **A1.** Thanks for this nice suggestion on improving the illustrations. We have carefully improved the figures in the revised version. Details are as follows:
>
> 1. Figure 1 has been enriched with legends and has been simplified to highlight the key components and major processes. The caption has also been improved to explain the whole process step by step. The usage of jargons are deliberately minimized to assist audience with various backgrounds. The full version of the figure with more details is put into Figure 8 of Appendix A.1.2.
>
> 2. Figure 2 has been replaced with an example of phase transition of a molecule system (e.g. water) between solid, liquid and gas.
>
>
> **Q2. For the baselines, the authors chose different continual learning framework as comparison. However for learning over multiple systems, there are some work [1,2] that directly learn a generalized neural simulators. It is suggested to also discuss their performance and show if continual learning is the better way to learning over different systems.**
>
> **A2.**
> Thanks for recommending these two excellent works. We have added detailed discussions on these works in Section 2.1 of the revised version. The discussion is also provided below.
>
> Generalized Graph Ordinary Differential Equations (GG-ODE) [1] focuses on systems with commonalities but exhibit different dynamics due to different environmental factors (exogenous factors). GG-ODE consists of two parts. One captures the commonalities to ensure generalizability while the other one captures the exogenous factors.
>
> High-order graph ODE (HOPE) [2] innovatively incorporates two types of high-order information, including information from high-order spatial neighborhood and high-order derivatives, into dynamical system modeling.
>
> Above all, GG-ODE [1], HOPE [2] and our MS-GODE target different perspectives of dynamical system learning. GG-ODE learns across systems governed by same physics laws in different environments, and aims at better model generalization. HOPE is an advanced way to better model the complex dynamics within in one system. While MS-GODE focuses on learning over systems governed by different laws, and aims at alleviating the forgetting issue.
>
> [1] Huang, Zijie, Yizhou Sun, and Wei Wang. "Generalizing graph ode for learning complex system dynamics across environments." Proceedings of the 29th ACM SIGKDD Conference on Knowledge Discovery and Data Mining. 2023.
>
> [2] Luo, Xiao, et al. "Hope: High-order graph ode for modeling interacting dynamics." International Conference on Machine Learning. PMLR, 2023.

---

> ### Author Response · Authors · 2024-11-20
> **Responses to Reviewer Pzyn (Part 2)**
>
> **Q3. For majority of the citation, it should be \citep (reference with a brackets) instead of \cite. For example, line 40-50.**
>
> **A3.** Thanks for reminding us about this issue. We have carefully checked and corrected all the citation forms in the revised paper.
>
> **Q4. For the experiment table 1, what does the results fine-tune and joint mean respectively? In line 400-402 the authors mention that for each baseline there are two options, so I overall do not get the meaning of the two rows in Table 1,which model are they based on ?**
>
> **A4.** Fine-tuning and Joint training are commonly used by existing continual learning works as the lower bound and upper bound on the performance. As introduced in 'Baselines \& model settings' of Section 4.2 (line 409-411 in the original submission) and Appendix A.3, Fine-tuning refers to the strategy that the backbone model is directly trained on a sequence of tasks without any continual learning technique. With fine-tuning, the catastrophic forgetting problem is fully exhibited. Joint-training refers to the strategy that the model is jointly trained on all tasks instead of sequentially trained on the tasks one by one. In other words, a jointly trained model does not follow the continual learning setting, and does not suffer from the forgetting problem.
>
> As introduced in the 'Model evaluation in CDL' part of Section 4.2 (line 396-397 in the original submission), we adopt LG-ODE as the backbone model for the continual learning techniques.

---

> ### Author Response · Authors · 2024-11-24
> **A Friendly Reminder to Reviewer Pzyn: Approaching Discussion Deadline**
>
> We sincerely thank the reviewer for the recognition of our work and providing valuable and constructive comments.
>
> We have made every effort to address all concerns raised, and have revised the paper wherever appropriate.
>
> With only three days remaining in the author-reviewer discussion period, we would greatly appreciate if the reviewer could let us know whether there are any remaining concerns.

---

> ### Author Response · Authors · 2024-11-29
> **We would really appreciate if the reviewer could let us know whether there are any remaining concerns**
>
> Dear reviewer,
>
> Again, we sincerely thank the reviewer for the recognition of our work and providing valuable and constructive comments to further improve our work.
>
> We have made every effort to address all concerns raised.
> Specifically, we have carefully revised Figure 1 and 2 as suggested by the reviewer, added discussion on the recommended related works, corrected the citation formats, and provided explanation on the baselines.
>
> With the approaching deadline of the author-reviewer discussion period, we would greatly appreciate if the reviewer could let us know whether there are any remaining concerns.
>
> Best regards,
>
> Authors

---

### Meta-Review · Area_Chair_pQdt · 2024-12-22

**Metareview:**

In this work, authors introduce Mode-switching Graph ODE (MS-GODE), a novel approach for Continual Dynamics Learning (CDL) that aims to model systems with evolving dynamics while preventing catastrophic forgetting. The work systematically investigates the CDL problem, proposes an innovative solution combining sub-network learning with mode-switching capabilities, and introduces a new benchmark dataset (Bio-CDL) for biological systems.

The paper demonstrates strong empirical results across different system configurations. The MS-GODE model leverages fixed backbone weights with adaptive binary masks for each dynamic mode, showing consistent performance improvements over baselines. The authors provide thorough ablation studies and theoretical analysis of why their approach better resolves the stability-plasticity dilemma in continual learning.

The primary strengths lie in: (1) The systematic formalization and investigation of the CDL problem, which has been largely overlooked despite its practical importance, as noted by Reviewer ZNCe. (2) The technical novelty of combining neural ODEs with mask-based continual learning, acknowledged by Reviewer VBaq. (3) The contribution of Bio-CDL benchmark, expanding beyond traditional physics-based systems, appreciated by all reviewers. (4) Strong empirical validation showing consistent performance improvements across diverse systems.

The work also have some limitations as follows. (1) The complexity of the multi-component architecture, though the authors provided detailed implementation guidelines addressing Reviewer Cscj's concerns. (2) Initial questions about technical novelty, which were adequately addressed in the rebuttal with detailed explanations of contributions. (3) Limited theoretical foundations, though this was addressed with formal analysis in the rebuttal to Reviewer ZNCe.

Altogether, the paper makes a clear contribution by identifying and systematically addressing an important problem in dynamics learning. The thorough empirical validation demonstrates consistent improvements across different systems. The paper will likely have meaningful impact on both theoretical understanding of continual learning for dynamic systems and practical applications in physics and biology domains. Accordingly, the manuscript is recommended for acceptance.

**Additional Comments On Reviewer Discussion:**

In addition to the points above, authors have comprehensively addressed reviewer concerns, particularly regarding technical novelty and theoretical foundations. The positive reception from all reviewers after rebuttal (scores of 5, 6, 6, and 8) indicates broad agreement on the paper's merit. While some limitations exist, they do not diminish the significant contributions made by this work. For future work, the authors may consider expanding to more domains beyond physics and biology, developing theoretical guarantees for performance under different dynamics shifts, and investigating scalability to larger systems. However, these suggestions are for future research directions rather than requirements for acceptance.

---

### Decision · Program_Chairs · 2025-01-22

Accept (Poster)